# Late developing cardiac lymphatic vasculature supports adult zebrafish heart function and regeneration

Michael RM Harrison[1]\*, Xidi Feng[1], Guqin Mo[1], Antonio Aguayo[1], Jessi Villafuerte[1,2], Tyler Yoshida[1,3], Caroline A Pearson[4], Stefan Schulte-Merker[5,6], Ching-Ling Lien[1,7,8]\*

[1]The Saban Research Institute of Children's Hospital Los Angeles, Los Angeles, United States; [2]Department of Biology, California State University of San Bernardino, San Bernardino, United States; [3]Department of Biological Sciences, Dornsife College of Letters, Arts and Sciences, University of Southern California, Los Angeles, United States; [4]Department of Neurobiology, Eli and Edythe Broad Center of Regenerative Medicine and Stem Cell Research, David Geffen School of Medicine, University of California, Los Angeles, Los Angeles, United States; [5]Institute of Cardiovascular Organogenesis and Regeneration, Faculty of Medicine, University of Münster, Münster, Germany; [6]CiM Cluster of Excellence (EXC1003 CiM), University of Münster, Münster, Germany; [7]Department of Surgery, Keck School of Medicine, University of Southern California, Los Angeles, United States; [8]Department of Biochemistry and Molecular Medicine, Keck School of Medicine, University of Southern California, Los Angeles, United States

**Abstract** The cardiac lymphatic vascular system and its potentially critical functions in heart patients have been largely underappreciated, in part due to a lack of experimentally accessible systems. We here demonstrate that cardiac lymphatic vessels develop in young adult zebrafish, using coronary arteries to guide their expansion down the ventricle. Mechanistically, we show that in *cxcr4a* mutants with defective coronary artery development, cardiac lymphatic vessels fail to expand onto the ventricle. In regenerating adult zebrafish hearts the lymphatic vasculature undergoes extensive lymphangiogenesis in response to a cryoinjury. A significant defect in reducing the scar size after cryoinjury is observed in zebrafish with impaired Vegfc/Vegfr3 signaling that fail to develop intact cardiac lymphatic vessels. These results suggest that the cardiac lymphatic system can influence the regenerative potential of the myocardium.

\*For correspondence:
mharrison@chla.usc.edu (MRMH);
clien@chla.usc.edu (C-LL)

**Competing interests:** The authors declare that no competing interests exist.

## Introduction

The cardiac vascular system is comprised of blood and lymphatic vessels. Arteries and connected capillaries transport oxygenated, nutrient-rich blood to the myocardium. Cardiac veins drain the blood back into the systemic circulation, and excess fluid and nutrients from the blood are hydrostatically released into tissue during this process. This interstitial fluid, immune cells, debris and waste products are then drained via the cardiac lymphatic vessels. The heart critically requires this continuous cycle through the myocardium for optimal function and interruption has pathological results including myocardial infarction (MI) or cardiac lymphedema (*Aspelund et al., 2016*; *Karunamuni, 2013*). Not only are the cardiac vessel systems required for heart function, but they are also likely to have active roles in disease resolution, at least helping to provide a permissive environment for regeneration (*Das et al., 2019*; *Klotz et al., 2015*; *Marín-Juez et al., 2016*). This has yet to be

**eLife digest** Human hearts have coronary vessels that supply oxygen and essential nutrients to the heart. When this supply is interrupted, a heart attack can occur. After a heart attack, scar tissue forms that impairs the heart's ability to pump blood around the body. The heart also has lymphatic vessels that drain excess fluid and remove waste products and damaged cells from the heart. Less is known about the lymphatic vessels and their role in heart disease.

Unlike human hearts, which scar easily, the zebrafish heart can regenerate after injury. Because of this, scientists often study zebrafish to try to find ways to improve healing of heart injuries in humans. However, it is not yet known whether lymphatic vessels contribute to regeneration of zebrafish hearts.

Now, Harrison et al. show that, in the zebrafish heart, lymphatic vessels develop after the coronary arteries. In fact, the coronary arteries provide a scaffold that the lymphatic vessels grow along.

When the zebrafish are genetically modified so that they lack coronary arteries, the lymphatic vessels fail to grow. Further experiments showed that, when the heart was injured by briefly freezing part of it, extra lymphatic vessels grew, but this did not happen when a part of the heart was removed via surgery. This may be because the cold-induced injury causes inflammation, which can trigger the growth of lymphatic vessels. The lymphatic vessels then help battle inflammation, allowing regeneration to proceed. Using genetically engineered zebrafish, Harrison et al. were then able to turn the genes that control lymphatic vessel growth on and off. They showed that zebrafish lacking lymphatic vessels in the heart are less efficient at regenerating heart tissue and develop more scar tissue after injury. This result is supported by the findings of a separate study conducted by Gancz et al.

The results suggest that stimulating the growth of lymphatic vessels or enhancing their activity in the injured heart may aid recovery. More studies may help scientists understand exactly how lymphatic vessels aid regeneration in zebrafish and whether promoting lymphatic vessel growth or activity may aid heart attack recovery in humans.

successfully leveraged clinically in part due to the lack of effective therapeutic strategies (*Taimeh et al., 2013*). A more detailed understanding of these systems, their formation, regeneration and promotion of positive disease environments are critically required.

Zebrafish represent a simple coronary vessel system, many features of which are conserved in the mammalian system. However, unlike mammals, the coronary structure in zebrafish starts to develop relatively late at juvenile stages when endocardial cells that give rise to coronary vessels sprout from the atrioventricular (AV) canal region of the heart to vascularize the ventricle (*Harrison et al., 2015*). The existence and development of a cardiac lymphatic system to complement this coronary system has yet to be described in zebrafish.

Extensive work using zebrafish embryonic models has yielded much of our understanding of embryonic lymphatic development (*Hogan and Schulte-Merker, 2017*). During trunk angiogenesis, lateral plate mesoderm-derived angioblasts from the ventral side of the posterior cardinal vein (PCV) migrate to the dorsal myoseptum, lose their connection with the PCV and become lymphangioblasts known as parachordal cells (*Hogan et al., 2009a*; *Nicenboim et al., 2015*). These cells then migrate from the midline along arteries under the guidance of Cxcl12b-Cxcr4a/b chemokine signaling to form lymphatic vessels, including the thoracic duct between the dorsal aorta and PCV (*Cha et al., 2012*). This thoracic duct is connected to a continuous dorsal longitudinal lymphatic vessel via intersegmental lymphatic vessels, which align along the same intersegmental arteries that provided guidance during their formation (*Bussmann et al., 2010*). This process of trunk lymphangiogenesis is completely blocked in embryos with mutations in genes encoding the secreted vascular endothelial growth factor C (*vegfc*) or its receptor *flt4/vegfr3* (*Hogan et al., 2009b*; *Villefranc et al., 2013*).

Vegfc is similarly required for the sprouting of lymphatic endothelial cells in mice (*Karkkainen et al., 2004*) and, as in zebrafish embryos, these lymphatic endothelial cells are derived primarily from venous endothelium (*Hägerling et al., 2013*; *Yang et al., 2012*) with some contribution from non-venous sources to specific lymphatic vasculature beds (*Ulvmar and Mäkinen, 2016*;

*Eng et al., 2019*). In the case of mouse cardiac lymphatic system, there is an additional contribution to the lymphatic vasculature from yolk sac endothelial cells (*Klotz et al., 2015*). Together with venous-derived lymphatic endothelial cells, they invade the embryonic heart at embryonic day (E) 12.5 from an extra-cardiac source to populate its base via the sinus venosus and outflow tract (*Flaht et al., 2012*; *Klotz et al., 2015*). This lymphatic endothelium then continues to sprout over the surface of the ventricle along Emcn-expressing cardiac veins. Vessels expressing lymphatic endothelial marker genes Lyve-1, Prox1 and Flt4 are identifiable specifically along the side of the cardiac veins by birth and continue expanding over the ventricle until postnatal (P) day 15 (*Klotz et al., 2015*). After a myocardial injury, the cardiac lymphatic vasculature is reactivated. Despite this expansion of the lymphatic network after myocardial infarction, the mammalian heart still scars rather than regenerates functional tissue. However, a reduction in this scarring can be observed when Vegfc is induced to further enhance lymphangiogenesis in injured adult mice (*Klotz et al., 2015*).

The cardiac lymphatic system is thought to regulate fluid homeostasis and provide immune system surveillance and clearance which may have important implications for cardiac tissue recovery after insult (*Vieira et al., 2018*). In contrast to the mammals, the zebrafish has retained a remarkable capacity to regenerate cardiac tissue after tissue damage or resection (*Gamba et al., 2014*; *Poss et al., 2002*). After resection or moderate cryoinjury to the apex or ventral wall of the ventricle, fully functional cardiac tissue is regenerated within 30–90 days and there is little or no detectable scar in the majority of injured zebrafish (*Chablais et al., 2011*; *González-Rosa and Mercader, 2012*; *Poss et al., 2002*). The regenerated tissue is also vascularized by blood vessels by this time and this vascularization supports the function of this regenerated tissue as well as the repair process itself (*Marín-Juez et al., 2016*).

Historical anatomical descriptions of lymphatic vessels surrounding the fish heart (*Hewson and Hunter, 1769*) suggest that these vessels constitute an ancient feature of jawed vertebrates, but the functional overlap of this vasculature with that of mammalian lymphatic vessels has been questioned (*Vogel and Claviez, 1981*). As such the existence and function of the cardiac lymphatic vasculature in zebrafish and its potential implication for normal tissue homeostasis as well as natural regeneration remains an open question of significant clinical interest.

We here characterize of the development of zebrafish cardiac lymphatic vasculature, describe its functions and analyze lymphangiogenesis during heart regeneration. Our results suggest that cardiac lymphatic vessels should be a promising therapeutic target for the treatment of heart disease and may help modulate a pro-regenerative immune response.

## Results

### Lymphatic vessels develop along coronary arteries of adult zebrafish

To begin to investigate the subtype identity of the cardiac lymphatic vasculature in the adult zebrafish, we used the *flt4:mCitrine* transgenic line that is expressed in both venous and lymphatic endothelial cells (*Gordon et al., 2013*; *van Impel et al., 2014*). *flt4:mCitrine* endothelial expression is detected at either systemic pole of the heart in the sinus venosus (SV) and on the surface of the bulbus arteriosus (BA) (*Figure 1A*). The population of endothelial cells on the outside of the BA was of particular interest given that they cover the entire BA, but do not extend down on the ventricle until adult stages (after 90dpf; *Figure 1A–C*). In young adult fish, *flt4:mCitrine*-positive cells bud from this BA population and appear to migrate down on the ventricle to form a vessel that extends to the apex of the ventricle (*Figure 1B–D*). To determine if this vasculature is venous or lymphatic we used the *prox1:Gal4-UAS:RFP* transgenic line that is not expressed in venous endothelium, but in lymphatic endothelial cells (LECs) and neurons (*van Impel et al., 2014*). Expression in the BA population suggests that this is lymphatic endothelium unlike the *flt4:mCitrine*-positive, *prox1:Gal4-UAS:RFP*-negative cells of the sinus venous, (*Figure 1E,F*, *Figure 1—figure supplement 1A, B*). The lack of markers exclusively specific to lymphatic endothelial cells has been one of the precluding factors for their visualization and study (*Jung et al., 2017*). We confirmed the lymphatic identity of this endothelium with four additional reporter lines known to be expressed in lymphatic endothelial cells: *lyve1b:GFP*, *lyve1b:DsRed* (*Okuda et al., 2012*), *stab1:YFP* (*Hogan et al., 2009a*) and *mrc1a:GFP* (*Jung et al., 2017*) (*Figure 1—figure supplement 1C-F*). As such the cardiac lymphatic system is identifiable in adult zebrafish and appears to extend from a BA population in older zebrafish. We

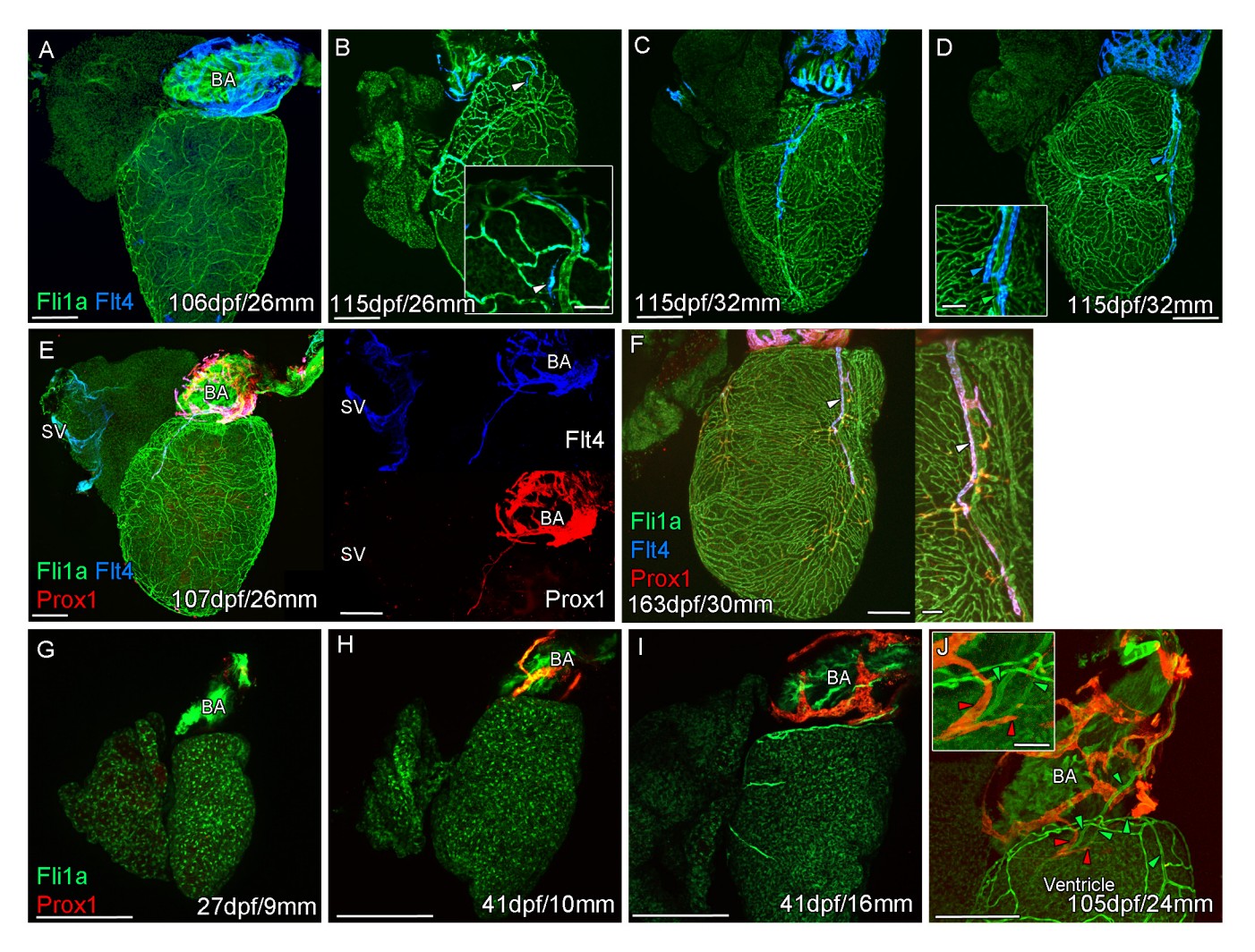

**Figure 1.** Prolonged development of the cardiac lymphatic system in juvenile and young adult zebrafish. Whole-mount confocal imaging of non-cleared (A-D, F-I) and cleared (E,J) adult (A-F) and juvenile (G-J) transgenic zebrafish hearts expressing the pan-endothelial marker fli1a:GFP (Green), venous/lymphatic endothelial marker flt4:mCitrine (blue: A-F) and lymphatic endothelial marker prox1:Gal4-UAS:RFP (red: E-J). (A) Through juvenile and young adult stages, *flt4:mCitrine*-expressing endothelial cells are restricted to the surface of the BA. (B) At 26 mm standard body length, sprouts from this BA population are first observed on the ventricle surface (arrowhead and insert). (C) These sprouts then extend down the ventricle and start to form a cardiac lymphatic vessel that eventually extends the entire length of the ventricle to the apex (D). The cardiac lymphatic vessel (D, inset, blue arrowhead) bridges the blood vessel (D, inset, green arrowhead) it accompanies. (E) Distinct from the venous endothelium of the SV, the BA population and the sprouts extending from it also express *prox1:Gal4-UAS:RFP*. (F) Both *flt4:mCitrine* and *prox1:Gal4-UAS:RFP* are maintained in the formed cardiac lymphatic vessel (arrowhead). (G) Larval and early juvenile zebrafish lack the *prox1:Gal4-UAS:RFP* expressing lymphatic endothelium on the BA. (H) This population is observed in juvenile zebrafish prior to the onset of and during (I) coronary vessel development. (J) In early adult stages, alignment of single cell sprouts off the lymphatic endothelium (red arrowhead) with blood vasculature (green arrowhead) is observed within the cleft between the BA and ventricle. Scale bars 200 μm (A-J) and 50 μm (insets, B, D, F), n ≥ 3 (A-J).

The online version of this article includes the following figure supplement(s) for figure 1:

**Figure supplement 1.** Expression of lymphatic endothelial marker genes in the cardiac lymphatic system.

next addressed the stage at which the BA population develops relative to coronary vessel development at juvenile stages.

The formation of a cardiac lymphatic vessel on the ventricle of adult zebrafish from a pre-existing population on the BA occurs very late and after the formation of the coronary vasculature in juvenile zebrafish. In rodent models, cardiac lymphatic vasculature on the ventricle is observed in embryos by E14.5 (*Klotz et al., 2015*). However in zebrafish we found that the lymphatic endothelium is first

established on the BA post-embryonically, before the development of the coronary vasculature (*Figure 1G–I*). The lymphatic endothelium extends to the base of the BA into the cleft between the BA and the ventricle (*Figure 1J*). Within this cleft the lymphatic endothelial cells sprout along coronary vessels to extend back up out on the ventricle side (*Figure 1J*). The sprouting lymphatic cells extend along the coronary arteries and track them as they progress down the ventricle towards the apex of the heart (*Figure 2A–F*). As the cells extend towards the apex they expand and flatten around the artery to form a vessel. This vessel is in close proximity to the arterial endothelium that expresses higher levels of *kdrl:mTurquoise* at this stage and shows more specific expression of *flt1^enh^:tdTomato*, *dll4:GFP* and *cxcr4a:mCitrine* (*Figure 2D–G*). This association between an artery

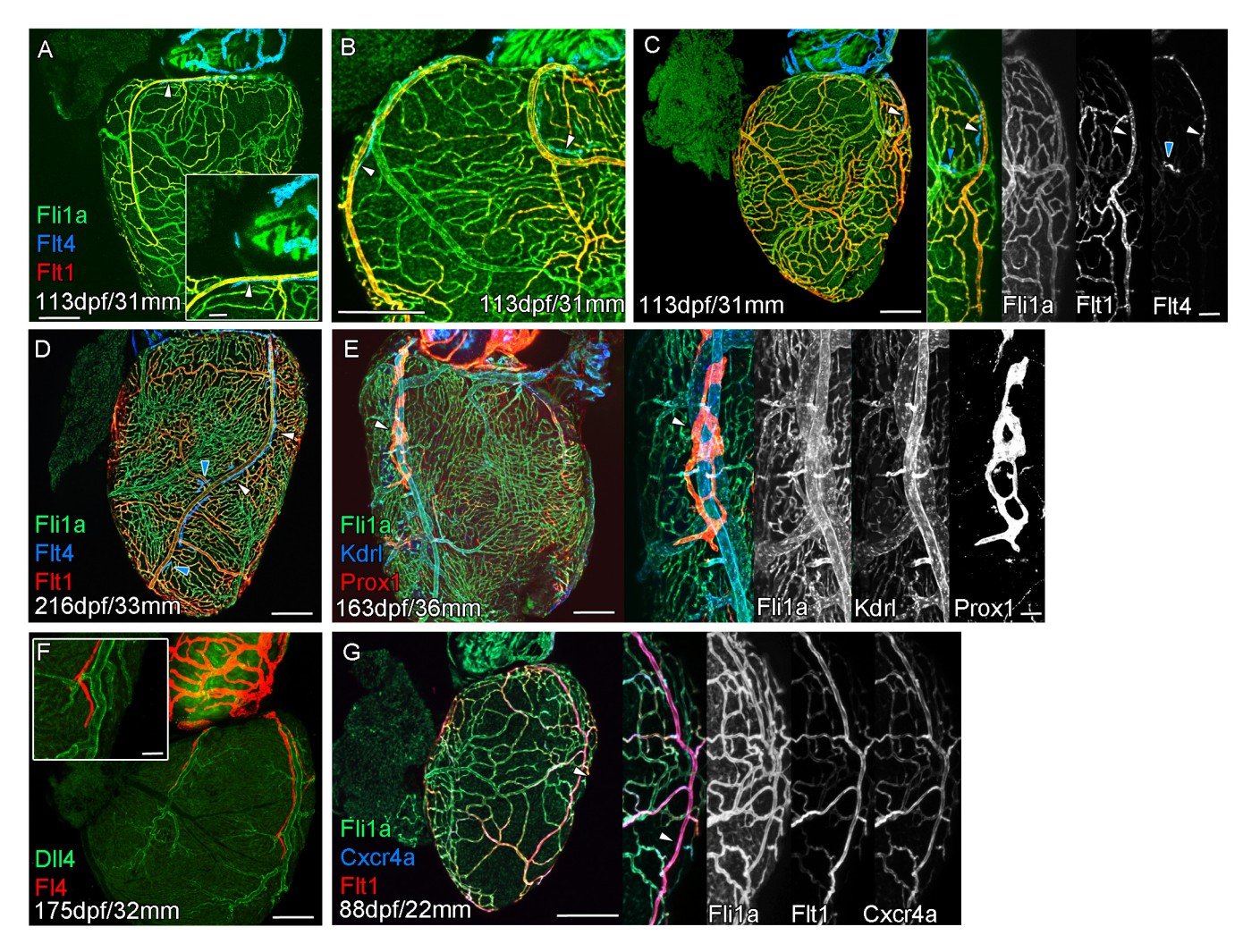

**Figure 2.** Extension of the cardiac lymphatic system along arteries in adult zebrafish. Whole-mount confocal imaging of adult transgenic zebrafish hearts expressing the pan-endothelial *fli1a:GFP* (green: **A-E, G**), venous/lymphatic endothelial marker *flt4:mCitrine* (blue: **A-D**, red: **F**), arterial endothelial cell marker *flt1^enh^:tdTomato* (red: **A-D, G**), lymphatic endothelial marker *prox1:Gal4-UAS:RFP* (red: **E**), arterially enriched blood endothelial marker *kdrl:mTurquoise* (blue: **E**) and arterial marker *dll4:GFP* (**F**). (**A-C**) *flt4:mCitrine*-positive sprouts migrate along *flt1^enh^:tdTomato* expressing arteries not *flt1^enh^:tdTomato*-negative veins (white arrowheads). The extension appears somewhat dynamic and fluid, with gaps or dissociations observed during extension and formation (blue arrowheads). This association continues as the cardiac lymphatic vessel forms on the ventricle (**D**). The formed vessel endothelium is *kdrl:mTurquoise*-negative and is in close proximity to high-*kdrl:mTurquoise* (**E**), *dll4:GFP* (**F**) expressing arteries. *flt1^enh^:tdTomato* expressing arteries also express high levels of *cxcr4a:mCitrene* in young adult zebrafish (**G**). Scale bars 200 μm (**A-G**) and 50 μm (insets **A, C, E, F and G**), n ≥ 3 (**A-G**).

The online version of this article includes the following figure supplement(s) for figure 2:

**Figure supplement 1.** Arterial association of extra-cardiac lymphatic vasculature.

and lymphatic vessel is also observed as the lymphatic vasculature extends along the aorta and branches off to the brachial aches. Here the lymphatic vessels associate with the brachial arteries and the arterial side of the gill filaments (*Figure 2—figure supplement 1A–D*).

## Development of a cardiac lymphatic system is incomplete in the absence of a coronary artery scaffold

In both zebrafish and mammals, there is a sequential development of the coronary vasculature and the cardiac lymphatic system suggesting that the coronary vessels provide a scaffold or guidance for the lymphatic system. To test this hypothesis, we analyzed the formation of cardiac lymphatic vessels in *cxcr4a* mutant zebrafish that lack coronary arteries. Unlike intersegmental lymphatic vessels, *cxcr4a:mCitrine* expression in the adult heart is restricted to the coronary arteries and is absent in migrating LECs and lymphatic vessels that form along them (*Figure 3A–C*).

Zebrafish with mutations in *cxcr4a* present with disrupted coronary vessel development and largely lack formed coronary vessels in young adult fish (*Figure 3—figure supplement 1B*). These mutants fail to regenerate after ventricular injury (*Harrison et al., 2015*). In older adult *cxcr4a* mutant zebrafish there is some formation of lumenized vessels, but the coronary vasculature structure remains highly disorganized over the ventricle. Often this coronary vasculature presents as a conglomerate of enlarged or dense interconnected by sparse intermediary non-luminized vessels or isolated cells (*Figure 3E and F*, *Figure 3—figure supplement 1A–F*). The density of this vasculature increases with time, as observed in wildtype transgenic zebrafish, however it lacks any identifiable coronary arteries (*Figure 3—figure supplement 1G–H*, *Figure 3—figure supplement 2C-M*). Analysis of lymphatic markers in *cxcr4a* mutant zebrafish demonstrated that cardiac lymphatic vessels do not extend down the ventricle in the absence of coronary arteries, but that the pre-forming BA population is unaffected (*Figure 3D–H*, *Figure 3—figure supplement 1A–H*, *Figure 3—figure supplement 2A-M*). There is a significant correlation between the extent of the coronary arterial tree over the ventricle and coverage of the ventricle by the cardiac lymphatic vessels (*Figure 3G*, *Figure 3—figure supplement 2M*), suggesting that the later uses the coronary vessels to migrate down the ventricle and that coronary artery derived signals may promote this process during development.

## The cardiac lymphatic system responds to injury of the ventricle

The zebrafish ventricle shows an amazing capacity to regenerate following insult or injury and previous work suggested that *vegfc* expression is elevated during this regeneration process (*Lien et al., 2006*). In adult hearts *vegfc* expression is observed on the BA in its junction with the ventricle base, but little detectable signal is observed on the ventricle itself (*Figure 4A*, *Figure 4—figure supplement 1A*). After amputation elevated *vegfc* levels are detectable at the wound site at 3dpa and 7dpa, but expression is reduced by 14dpa and absent thereafter (*Figure 4B*, *Figure 4—figure supplement 1B-G*). To test the effect of having remaining necrotic tissue and a pre-existing extracellular matrix after injury, we used the more severe cyroinjury model. Cryoinjury also results in *vegfc* expression at 3dpc also, however subsequent expression appears to be more prolonged and remains elevated after 42dpc (*Figure 4C and D*, *Figure 4—figure supplement 1H-L and G*). Following resection of the ventricle apex there is little expansion of the cardiac lymphatic vasculature, with only some hearts showing extension of the lymphatic vessels into the wound site after 14dpa (*Figure 4E–H*). If present these lymphatic vessels appear to largely be extensions of the existing lymphatic vessel into the apex and do not appear to be enlarged or expanded (*Figure 4E–H,Q and R*; 4/7 hearts at 60dpa show wound site lymphatic vessels). In contrast, after cryoinjury migration of LECs into the wound site is observed at 14dpc, but also enlargement and expansion of the vessel on the ventricle (*Figure 4I–P,S–V*). Particularly at later stages, a highly branched network of cardiac lymphatic vasculature is observed within the wound site (*Figure 4L–P,S–V*). These regenerated lymphatic vessels express *flt4:mCitrine*, *prox1:Gal4-UAS:RFP*, *lyve1:RFP*, *mrc1a:eGFP* (*Figure 4L,N–P*). Quantification of lymphatic vessel coverage within a set area of the ventricle shows significantly more lymphatic vessels in or around the wound site after cryoinjury when compared to the resection model (*Figure 4U*). In addition post-cryo-injured hearts have a more branched cardiac lymphatic structure over the ventricle (*Figure 4V*). This suggests that the presence of necrotic tissue and scar formation promotes the formation of more lymphatic vessels after cardiac injury. We next wanted to

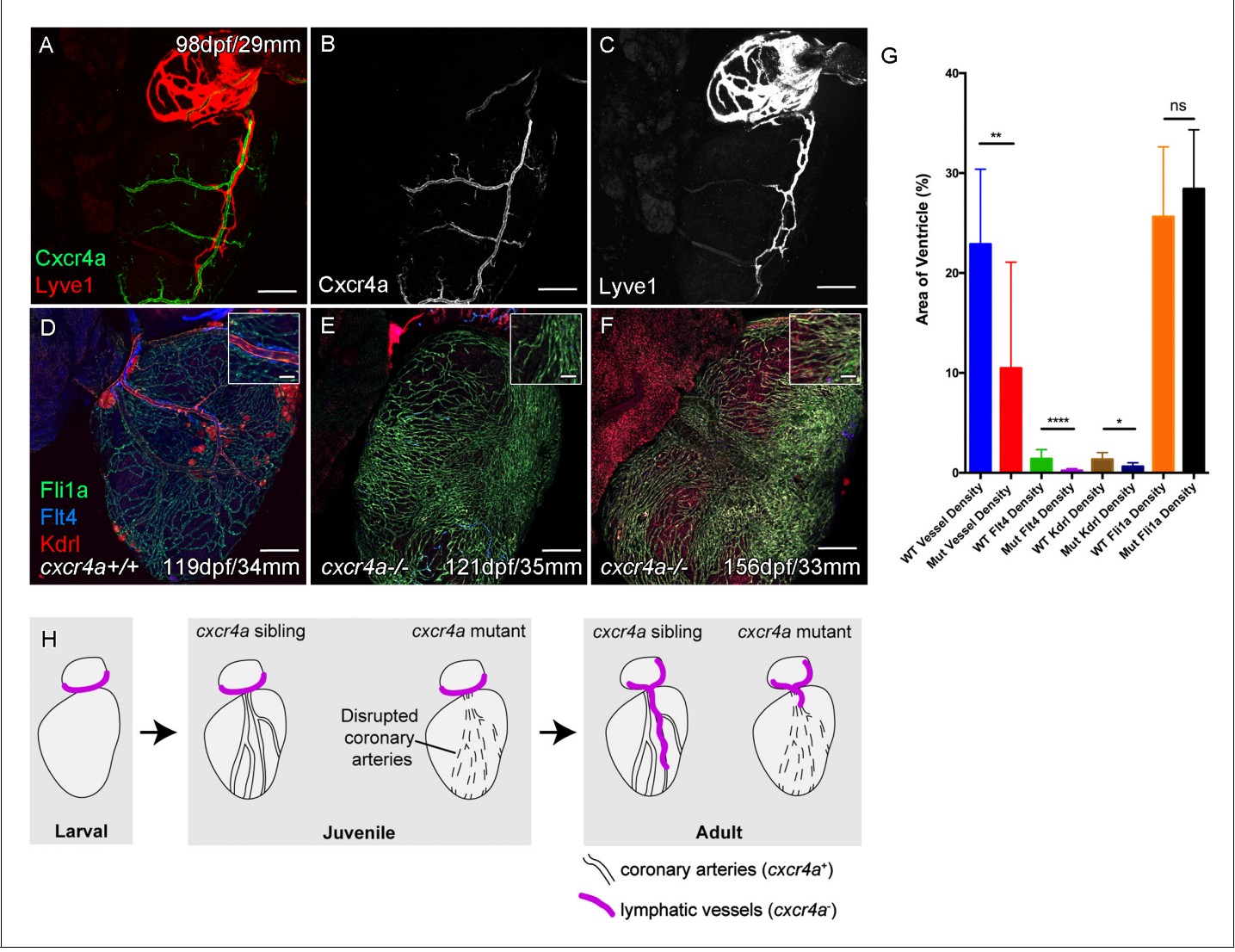

**Figure 3.** Cardiac lymphatic development requires a coronary artery scaffold. Whole-mount confocal imaging of adult transgenic zebrafish hearts expressing arterial marker *cxcr4a:mCitrine* (green: **A and B**), and lymphatic endothelial markers *lyve1:RFP* (Red: **A and C**) and *flt4:mCitrine* (Blue: **D-F**), the pan-endothelial marker *fli1a:GFP* (green: **D-F**), and the arterially-enriched pan-endothelial marker *kdrl:mTurquoise* (red: **D-F**). (**A-C**, n = 8) *cxcr4a: mCitrine* expression is restricted to the coronary arteries and not detectable in *lyve1:RFP*-positive LECs. (**D**) Cardiac lymphatic vessels in control (*cxcr4a +/+*, n = 8) zebrafish. (**E and F**) In *cxcr4a* mutants (*cxcr4a-/-*, n = 12) that have abnormal and discontinuous coronary vasculature the cardiac lymphatic extension is stunted. (**G**) Graph showing significant reductions in vessel density, *kdrl* expression in blood vessels and *flt4* expression on the surface of the *cxcr4a-/-* ventricle. There is comparable levels of *fli1a*-expressing cells on the surface of the ventricle between larger/older *cxcr4a-/-* and control zebrafish, but loss of the coronary structure. (**H**) Schematic representation of the regulation of cardiac lymphatic developmental guidance by coronary arteries. Scale bars 200 μm.

The online version of this article includes the following source data and figure supplement(s) for figure 3:

**Source data 1.** Source data for (**G**).
**Figure supplement 1.** Cardiac lymphatic vessels extension is correlated with extent of the coronary arterial tree.
**Figure supplement 2.** Cardiac lymphatic vessels extension is correlated with extent of the coronary arterial tree.

address if this was related to the function of the cardiac lymphatic system and its potential utilization in a disease setting.

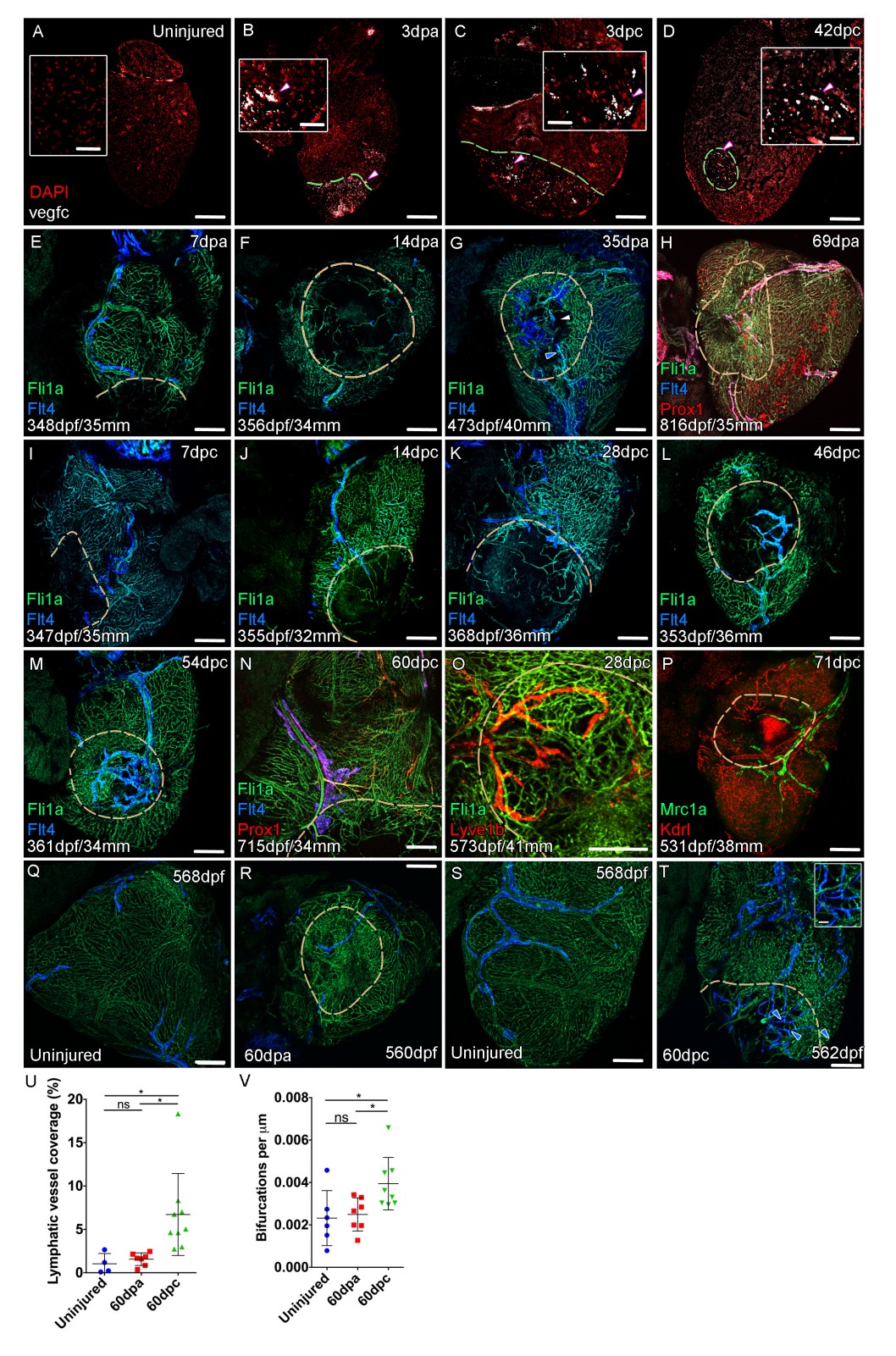

**Figure 4.** Lymphangiogenesis in response to injury of the zebrafish ventricle. Confocal images of RNAScope section in situ hybridization to detect *vegfc* (white: A-D). Whole-mount confocal imaging of adult transgenic zebrafish hearts expressing the pan-endothelial marker *fli1a:GFP* (Green: E-O, Q-T) and lymphatic endothelial markers *flt4:mCitrine* (blue: E-O, Q-T), *prox1:Gal4-UAS:RFP* (red: H and N), *lyve1:RFP* (Red: O), *mrc1a:eGFP* (Green: P) and endothelial marker *kdrl:mCherry* (Red: P). (E-H) Response to amputation of *flt4*-expressing lymphatic vasculature during regeneration at 7 (E), 14 (F), 35

*Figure 4 continued on next page*

*Figure 4 continued*

(**G**) and 69dpa (**H**). Lymphatic vasculature is observed to extent into the wound site at 35dpa to connect to the vessel on the BA (blue arrowhead) with some disconnections observed (white arrowhead). Response of *flt4*-expressing lymphatic vasculature to cryoinjury at 7 (**I**), 14 (**J**), 28 (**K**), 46 (**L**), 54 (**M**) and 60dpc (**N**). Expression prolife of regenerated lymphatic vasculature shows expression of *prox1:Gal4-UAS:RFP* (red: **H** and **N**), *lyve1:RFP* (Red: **O**) and *mrc1a:eGFP* (Green: **P**). (**Q** and **S**) Lymphatic vessels (blue arrowheads) extend to the apex in uninjured zebrafish. After amputation, there is little lymphangiogenesis and most of the regenerated vasculature lacks lymphatic marker expression with some minor extension into the wound site (**R**, blue arrowhead). Extensive lymphangiogenesis into and around the wound site (60dpc, **T**, blue arrowheads) is stimulated following damage to the myocardium with cryoinjury. (**U**) Quantification of the increase in area of *flt4:mCitrine* expression within a set 600 $\mu m^2$ region centered on the wound site of sibling zebrafish subject to severe cryoinjury (n = 9) in comparison to that resulting from amputation (n = 7; *p=0.0132, unpaired t-test) or the uninjured apex (n = 4; *p=0.0407, unpaired t-test). (**V**) Quantification of increased branching of lymphatic vessels on the ventricle in response to cryoinjury (n = 9) in comparison to uninjured hearts uninjured hearts (n = 6; *p=0.0293, unpaired t-test) and those subjected to amputation (n = 7; *p=0.0140, unpaired t-test). Scale bars 200 $\mu m$ (**A-D**) and 50 $\mu m$ (insets).

The online version of this article includes the following source data and figure supplement(s) for figure 4:

**Source data 1.** Source data for (**U**).
**Source data 2.** Source data for (**V**).
**Figure supplement 1.** *vegfc* expression after cardiac injury.
**Figure supplement 1—source data 1.** Source data for Supplement 1(**G**).

## Cardiac lymphatic vasculature functionally supports the heart during regeneration and homeostasis

The lymphatic system in mammals is known to maintain fluid homeostasis and modulate immune surveillance and clearance (***Alitalo, 2011***). Given these functions it has long been considered an integral apparatus in the maintenance of heart function and postulated to encourage a regenerative response to injury (***Aspelund et al., 2016***; ***Karunamuni, 2013***; ***Klotz et al., 2015***; ***Vieira et al., 2018***). In order to ascertain the function of the zebrafish cardiac lymphatic vasculature and in light of the discontinuous nature of LECs sometimes observed in these vessels (***Figures 2*** and ***4***), we established an intra-myocardial injection assay of microspheres (MS) and quantum dots (Qdots). Within 1 hr both were found at the injection site (***Figure 5A***), however the smaller Qdots (<10 nm diameter) were more dispersed from the injection site as well as being concentrated within the lymphatic lumen (***Figure 5A–C***, ***Figure 5—video 1***). This demonstrates that, despite the discontinuations in *lyve1:RFP* marker expression, the ventricular cardiac lymphatic vessel forms a blunt ended contiguous tube alongside the blood vessels. In addition these vessels are pervious to the uptake of fluid and small, but not large, debris as the larger microspheres (200 nm diameter) remained at the injection site, but the smaller Qdots become dispersed within the interstitium and are taken up into the lymphatic vessels.

In the uninjured heart, resident macrophages (labeled with IB4) were observed near the vasculature and there were very few neutrophils (Mpx-positive) present on the ventricle (***Figure 5—figure supplement 1A***). Given the strong lymphangiogenic response to cryoinjury (***Figure 4***), we decided to investigate the immunological role of the cardiac lymphatic system after a mild cryoinjury. Following injury, macrophages migrate to the wound site and are joined by a large number of neutrophil cells (***Figure 5—figure supplement 1B***). Clearance of these cells is known to be a key step in the process of regenerating heart tissue (***Lai et al., 2017***; ***Vieira et al., 2018***). We find lymphatic vessels may provide a conduit for this clearance during zebrafish heart regeneration. After 1 and 7dpc (mild cryoinjury) mpx-positive neutrophil cells and debris became highly enriched within or on the cardiac lymphatic vessels on the BA, a location relatively distant from the more apex ventricular wound site (***Figure 5D–H***, ***Figure 5—figure supplement 1C–G***). Despite weak expression on the ventricle of *lyve1:GFP* that precluded antibody detection above background, Mpx-positive cells were observed to align up along the optically dense blood vessels on the ventricle (***Figure 5E*** insert) consistent with their association and uptake into lymphatic vessels. Immune cells were observed within the labeled BA cardiac lymphatic vessels after injury (***Figure 5G***, ***Figure 5—figure supplement 1E and F***, ***Figure 5—videos 2–5***). This suggests these cells can be taken up by the cardiac lymphatic vasculature at or near the wound site after injury and then transported away from the heart via the lymphatic vasculature on the aorta.

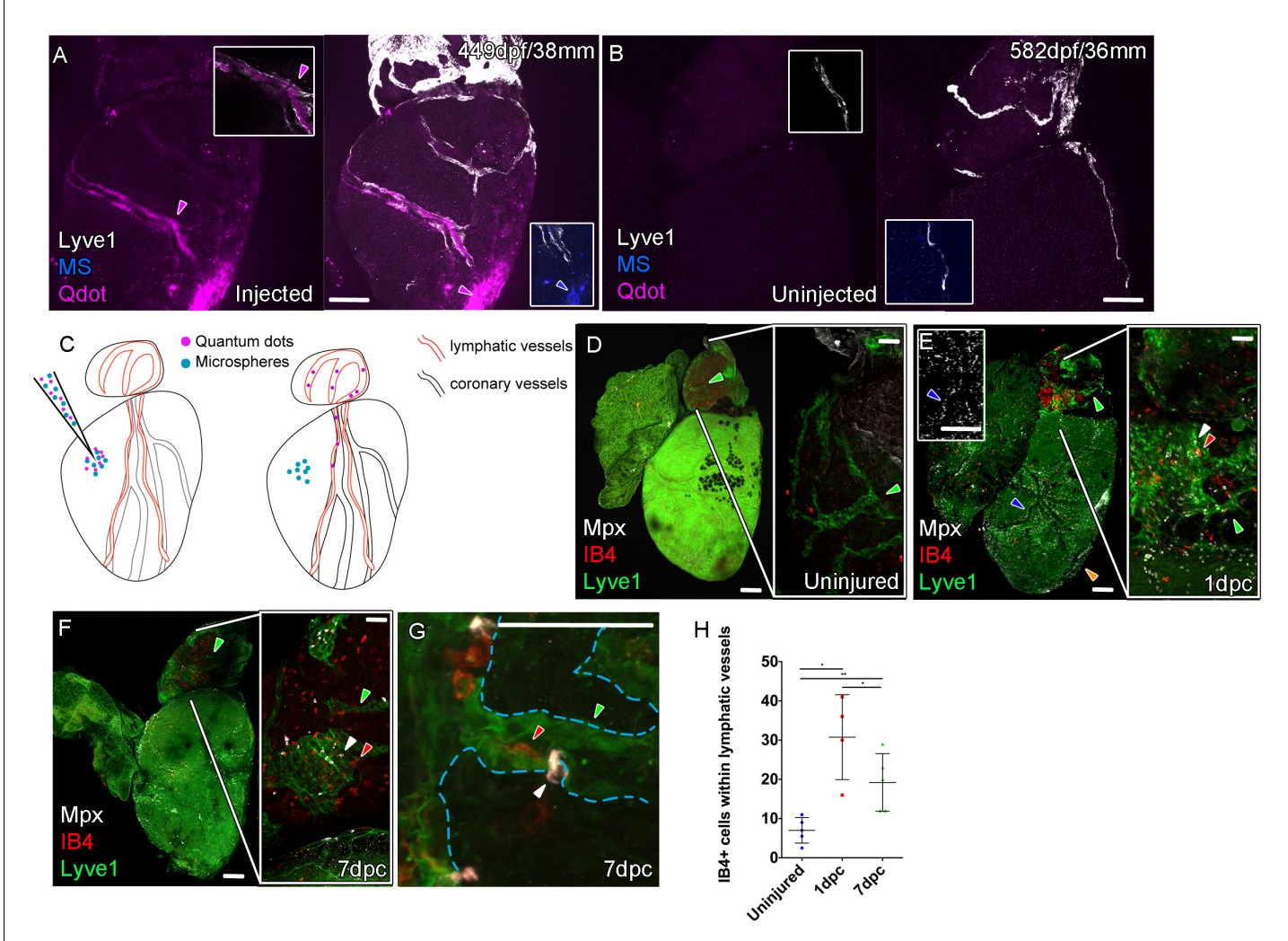

**Figure 5.** Function of the cardiac lymphatic vasculature in regeneration. Whole-mount confocal imaging of adult transgenic zebrafish hearts expressing *lyve1:RFP* (White: **A and B**) *and lyve1:GFP* (Green: **D-G**) to mark lymphatic vessels. Additional fluorescence signal from injected Firefli Fluorescent Blue-dyed Microspheres (MA; blue: **A and B**), Quantum dots (Qdot; magenta: **A and B**) and immuno-labeled neutrophils (Mpx; white: **D-G**) and Macrophage (IB4; red:**D-G**). (**A**) Intra-myocardial injection of 200 nm microspheres (cyan) and 10 nm Qdots (magenta) results in both being deposited at the injection site (magenta/cyan arrowheads). Qdots, but not MS are also highly concentrated within the lymphatic vessels within the *lyve1:RFP* positive endothelium (arrowheads, **A**; uninjected control, **B**). (**C**) Schematic representation of nanoparticle uptake by the cardiac lymphatic system. (**D**, n = 5) Uninjured zebrafish heart labeled with antibodies against GFP (lyve1:GFP) and Mpx and lectinB4 (IB4), showing relatively few Mpx-positive neutrophils or IB4-positive macrophage on the ventricle or BA, including within the lymphatic vessels on the BA (green arrowheads). (**E**, n = 4) 1 day post-cryo-injured (dpc) heart (mild injury) labeled at the same time shows a massive increase in Mpx-expressing neutrophils and more moderate increases IB4-positive macrophage including within the lymphatic vasculature (green arrowhead (GFP) white arrowhead (Mpx) red arrowhead (IB4). Mpx-positive cells are concentrated along the visibly darker blood vessels (inset, blue arrowhead) away from the wound site (orange arrowhead). (**F**, n = 5) 7dpc heart showing persistent elevation of Mpx-positive and IB4-positive cells on the ventricle and BA, where they are associated with lymphatic vessels. (**G**) High resolution imaging of *lyve1:GFP* -positive vessels (green arrowhead, outlined in blue) on the BA of a 7dpc heart IB4-positive macrophage (red arrowhead) within the vessel and Mpx-positive neutrophil (white arrowhead) on the surface of the vessel (**H**) Graph showing quantification of increase in IB4-positive cells within the BA lymphatic vessels of 1dpc (n = 4) and 7dpc (n = 5) zebrafish hearts compared to those of uninjured hearts (n = 5). *p<0.05; **p<0.01, unpaired t-test, Mean and SD. Scale bars 200 µm (**A-G**) and 50 µm (right panels, **D-F**).

The online version of this article includes the following video, source data, and figure supplement(s) for figure 5:

**Source data 1.** Source data for (**H**) and Supplement 1(**G**).

**Figure supplement 1.** Immune response by macrophage and neutrophils to mild cyro-injury Uninjured zebrafish heart.

**Figure 5—video 1.** Concentrated Qdot signal within the cardiac lymphatic lumen after intermyocardial injection.

https://elifesciences.org/articles/42762#fig5video1

**Figure 5—video 2.** IB4-positive macrophage within the a cardiac lymphatic vessel.

*Figure 5 continued on next page*

## Vegfc promotes the formation of cardiac lymphatic vasculature after the initiation of coronary vessel development

Given Vegfc's well established role in lymphatic vasculature development (*Aspelund et al., 2016*; *Joukov et al., 1996*; *Kaipainen et al., 1995*) and its potential role after injury (*Figure 4*) we decided to test if it was utilized during cardiac lymphatic system development. We inactivated the Vegfc ligands by induced expression of a soluble version of the Flt4/Vegfr3 receptor (serving as a ligand trap; sFlt4) in transgenic juvenile zebrafish during the formation of the ventricular cardiac vessels (*Matsuoka et al., 2017*). Unexpectedly, in heat shocked non-transgenic control zebrafish we observed an increase in lymphatic vessels on the heart relative to non-heat shocked controls (*Figure 6A,B*, *Figure 6—figure supplement 1A,B,D*). This increase in cardiac lymphatic vessels is not attributable to an increase in body size, in fact, heat shocked zebrafish showed a slight reduction in the overall length compared to non-heat shocked zebrafish at the same stage (*Figure 6—figure supplement 1E*). In stark contrast, the induced transgenic zebrafish that expressed sFlt4 from 35dpf had few or no LECs on the ventricle and no cardiac lymphatic vessels on the ventricle (*Figure 6C and D*, *Figure 6—figure supplement 1C,D*). This was also the case with sFlt4 induction during and after coronary vessel induction in older juvenile (from 71dpf) and adult zebrafish respectively (from 91dpf; *Figure 6—figure supplement 1G–J*). Induction of sFlt4 from 35, 71 or 92dpf did not overtly affect the development of the coronary vasculature, nor did we observe overt edema or malformation (*Figure 6B,C*, *Figure 6—figure supplement 1B, C, F, G-N*). Ventricle tissue appears to be normal with sFlt4 induction in juvenile zebrafish and does not significantly affect cardiomyocyte (CM) or non-CM numbers or the overall size of the ventricle (*Figure 6—figure supplement 1O-S*). This suggests that functional post-embryonic Vegfc signaling is specifically critical for the formation of this relatively late developing structure. We next used these zebrafish that lacked cardiac lymphatic vessels in the presence of coronary arteries to specifically test the effect of this system on cardiac regeneration.

## Lack of cardiac lymphatic vasculature results in persistence of scar tissue following cryoinjury of the ventricle

To test if the zebrafish cardiac lymphatic system has the potential to influence the varying levels of fibrosis observed after cardiac injury (*Lai et al., 2017*), we used zebrafish with differing levels of cardiac lymphatic vasculature due to the developmental heat shock and induction of a sFlt4 receptor (*Figure 6A–D*), and performed cryoinjury and assayed scar size 60 days later without post injury induction of sFlt4 (60dpc, *Figure 6D*). Scoring these scars on the basis of severity we found large deposits of scar tissue in the transgenic fish (5/5, large scar,>0.005 mm$^3$), in comparison to that of non-transgenic siblings, where the majority of scarring was relatively small (3/5, minimal scar,<0.005 mm$^3$) (*Figure 6E and F*). Comparison of heart tissue scaring levels shows a significant increase in scarring severity with a reduction in the cardiac lymphatic vasculature (*Figure 6H*). Hearts lacking ventricular cardiac lymphatic vessels have a significantly larger scar volume (*Figure 6—figure supplement 2A*). This increase in scar volume could not be attributed to an observable defect in the coronary vasculature or a loss of coronary vessel angiogenesis at the wound site (*Figure 6—figure supplement 1A-C, G-J*; *Figure 6—figure supplement 2B-E*).

To test if the induction of *vegfc* after injury supported this function we inactivated the Vegfc ligands by induced expression of sFlt4/Vegfr3 receptor 2 days prior to and then after cryoinjury (*Figure 6—figure supplement 3A*). As is the case with induction of sFlt4 during cardiac lymphatic development, we observed only moderate non-significant effects on contrary vessel regeneration

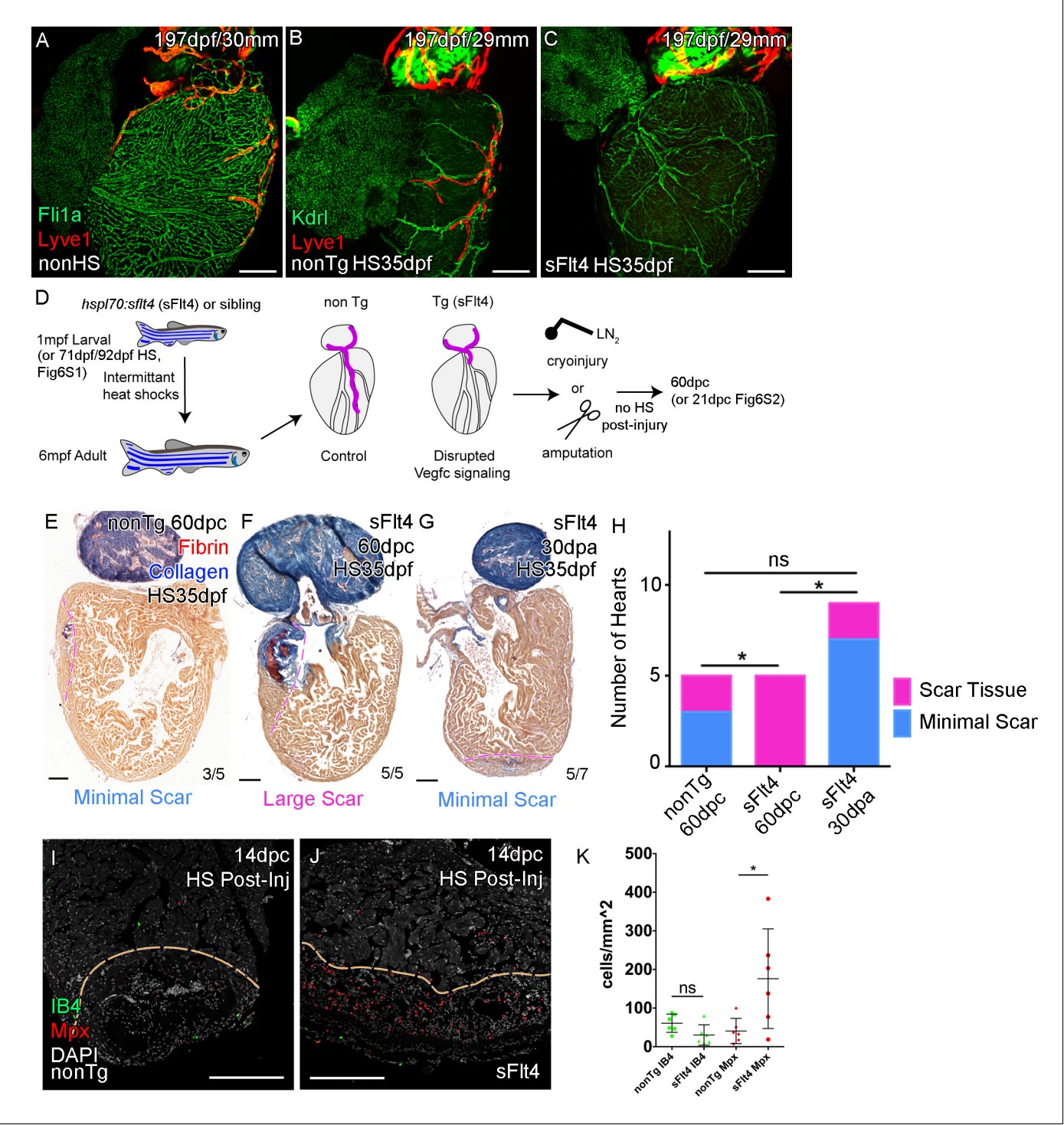

**Figure 6.** Cardiac lymphatic development requires Vegfc signaling and helps reduce scar volume in response to cryoinjury. Whole-mount confocal imaging of adult transgenic zebrafish hearts expressing the pan-endothelial *fli1a:GFP* (green: A), Arterially and endocardially enriched endothelial marker *kdrl:GFP* (green: B, C), lymphatic endothelial marker *lyve1:RFP* (Red: A-C). (A) Heart with normal cardiac lymphatic vessels from stage and size-matched control zebrafish (nonHS, n = 7). (B) Hearts from non-Transgenic zebrafish (nonTg, n = 3) that are exposed to heat shock with cardiac lymphatic vessels on the heart that extend further and cover more of the ventricle. (C) In contrast, hearts from *hsp70l:sflt4* zebrafish (sFlt4, n = 7), heat shocked in the same way lack extension of the cardiac vasculature onto the ventricle. (D) Schematic representation of the experimental procedure and regulation of cardiac lymphatic development by inducing sFlt4. AFOG-histology images showing fibrin in red and collagen in blue in developmentally heat shocked (1 to 6mpf) non-transgenic (nonTg; E) and transgenic (*hsp70l:sflt4*; sFlt4; F and G) zebrafish 60 days following severe cryoinjury (60dpc, 8mpf, E and F) or 30 days after apex resection (30dpa, 7mpf, G). Scar volume was estimated and grouped based on severity for cryoinjured hearts

*Figure 6 continued on next page*

*Figure 6 continued*

(minimal scar,<0.005 mm$^3$, **E**; large scar,>0.005 mm$^3$, **F**) and amputated hearts (minimal scar, **G**). (**H**) Graph showing the distribution of tissue scaring between non-transgenic (nonTg, n = 5) and transgenic (*hsp70l:sflt4*) zebrafish at 60dpc (sFlt4, n = 5) and 30dpa (n = 7), *p<0.05, two-tailed Chi-Squared. Fluorescence signal from immuno-labeled neutrophils (Mpx; Red: **I** and **J**) and Macrophage (IB4; green: **I** and **J**) in and near the 14dpc woundsite (demarked orange; Post-injury sFlt4 induction, n = 6 per group). (**J**) Elevated level of Mpx-positive immune cells at the 14dpc woundsite of fish with post-injury induced sFlt4 (n = 6). (**K**) Separate quantification of Mpx and IB4-positive cells in and within 100 μm of the 14dpc woundsite (average of at least three images through woundsite, individually normalized to woundsite area, n = 6 per group). *p<0.05; **p<0.01; ***p<0.001, unpaired t-test, Mean and SD. Scale bars 200 μm.

The online version of this article includes the following source data and figure supplement(s) for figure 6:

**Figure supplement 1.** Juvenile Vegfc signaling is required for cardiac lymphatic development.
**Figure supplement 1—source data 1.** Source data for Supplement 1(D).
**Figure supplement 1—source data 2.** Source data for Supplement 1(F).
**Figure supplement 1—source data 3.** Source data for Supplement 1(Q–S).
**Figure supplement 1—source data 4.** Source data for Supplement 1(J and K).
**Figure supplement 2.** Blocking juvenile cardiac lymphatic development results increased scaring after cryoinjury at adult stages.
**Figure supplement 2—source data 1.** Source data for Supplement 2(D) and (E).
**Figure supplement 2—source data 2.** Source data for (H) and Supplement 2(A).
**Figure supplement 2—source data 3.** Source data for (K), Supplement 2(H) and Supplement 4(C).
**Figure supplement 3.** Induction of sFlt4 after cryoinjury has no significant effect on coronary vessels or heart morphology.
**Figure supplement 3—source data 1.** Source data for Supplement 3(F) and (G).
**Figure supplement 4.** Post-injury induction of sFlt4 limits myocardial wall expansion and scar internalization.
**Figure supplement 4—source data 1.** Source data for Supplement 4(H-J).
**Figure supplement 4—source data 2.** Source data for Supplement 4(M).
**Figure supplement 4—source data 3.** Source data for Supplement 4(P).
**Figure supplement 5.** Loss of lymphatic vessels does not effect scar formation after amputation injury.

(*Figure 6—figure supplement 3B-G*) and no relative difference in post-injury heart morphology or size with sFlt4 post-injury induction (*Figure 6—figure supplement 3H-K*). Despite the lack of significant extension into the wound site at this stage, an increase in Mpx-positive neutrophils was observed at the 14dpc wound site with post injury induction of sFlt4 (*Figure 6I–K*), suggesting that blocking Vegfc-signal could prolong the inflammatory response, potentially by limiting the removal of Mpx-positive cells and debris from the wound site. Indeed, some sFlt4-induced hearts still showed elevated neutrophil levels at later stages of regeneration (Developmental induction, *Figure 6—figure supplement 2F-H*; Post-injury induction, *Figure 6—figure supplement 4A-C*).

Post-injury induction to block the response of the cardiac lymphatic vasculature did result in a reduction in the regeneration of the cardiac wall and resulting internalization of the scar tissue (*Figure 6—figure supplement 4D-J*). However the scar area itself showed no significant increase in these zebrafish (*Figure 6—figure supplement 4D-J*). There appears to be no inhibition of the induction of cardiomyocyte proliferation or epicaridal activation following injury with overexpression of sFlt4 (*Figure 6—figure supplement 4K-P*), suggesting instead that during regeneration the lack of ventricular cardiac lymphatic vessels, or perturbing their response to injury, shifts the balance of the wound site from pro-regenerative to pro-fibrotic. Consistent with this hypothesis we observed an inability of zebrafish that lack cardiac lymphatic vessels to modulate the wound environment and to resolve scar specifically after cryoinjury (*Figure 6H*). The majority of amputation-injured heat shocked transgenic *hsp70l:sflt4* zebrafish had only minimal scaring after 30 days (30dpa, *Figure 6G and H*). The amount of scar tissue in those zebrafish with detectable scars at 30dpa was relatively small and also observed in amputated control zebrafish, suggesting the role of cardiac lymphatic vessels is less critical for regeneration following resection (*Figure 6—figure supplement 5A–G*).

## Discussion

We here show that during zebrafish post-embryonic stages, there is a sequential development of blood vessel network and the cardiac lymphatic system as occurs embryonically in mice (*Flaht et al., 2012; Klotz et al., 2015*). The juvenile zebrafish heart undergoes significant morphological change and expansion due to the physiological demands of increasing body size on heart output

(*Gupta and Poss, 2012*; *Harrison et al., 2015*). This expanded myocardium likely requires auxiliary oxygenated blood supply and lymph clearance to function optimally (*Harrison et al., 2015*; *Figure 1*). By contrast, in mammals the evolution of the pulmonary system has permitted a much elevated systemic blood pressure and metabolic rate that requires a greater cardiac output from birth (*Bettex et al., 2014*). As such, the needs of the post-natal compact myocardium have driven the timing of cardiac blood and lymphatic vasculature development embryonically.

The sequential development of cardiac blood and lymphatic vasculature is conserved between mammals and fish and has also been reported within the embryonic zebrafish trunk (*Bussmann et al., 2010*). However, as in the zebrafish heart, intersegmental arteries, not veins, provide a scaffold for the extension of lymphatic vessel sprouts and are required for their formation (*Bussmann et al., 2010*). We have identified a population of LECs that migrate down the arteries in the zebrafish heart (*Figures 1–3*) and a similar interaction of developing cardiac lymphatic vessels with arteries has been described in the human fetus (*Kampmeier, 1928*). This is in contrast to mice in which the cardiac lymphatic vessels track cardiac veins rather than coronary arteries (*Klotz et al., 2015*). In humans, like zebrafish, large arteries (and to lesser extent veins) are located subepicardially; in contrast the mouse heart has subepicardial veins, but deeper intramyocardial arteries (*Sharma et al., 2017*). Zebrafish lymphatic development may represent the ancestral mechanism of lymphatic guidance which has been entirely conserved or diverged to different degrees across the mammalian class (*Ratajska et al., 2014*). In each case, the cardiac lymphatic vessels appear to follow the more superficial blood vasculature on the ventricle and our data suggest this is required for or promotes their extension over the heart.

We have demonstrated that loss of the coronary artery scaffold results in a failure of these vessels to extend down the ventricle in *cxcr4a* mutants that lack coronary arteries. Although Cxcr4-Cxcl12 signaling has been implicated in the arterial association of LECs in the zebrafish trunk (*Cha et al., 2012*), we do not observe *cxcr4a:mCitrine* expression in lymphatic endothelial cells, but do not rule out that a similar direct signaling pathway, for example through Cxcr4b, could also occur in the cardiac system and that this may be compromised in *cxcr4a* mutants.

We have however identified a signaling pathway required for the development of the cardiac lymphatic system in zebrafish. Vegfc/Flt4 signaling has been shown to have important roles in embryonic and perinatal angiogenesis (*Hogan et al., 2009a*), lymphangiogenesis, (*Küchler et al., 2006*; *Nurmi et al., 2015*; *Yaniv et al., 2006*) and maintenance of mature intestinal lymphatic vessels (*Nurmi et al., 2015*; *Tammela et al., 2008*). We here show that Vegfc is critical for the outgrowth of cardiac lymphatic vessels along the arteries of the ventricles in adult zebrafish (*Figure 6*).

Our experiments also demonstrate that regular heat shock treatment of zebrafish actually promotes cardiac lymphatic development in non-transgenic control zebrafish (*Figure 6*). Such positive effect on lymphatic development is likely due to the induced stress response and associated demand on cardiac output that is observed during heat shock (*Sallin and Jaźwińska, 2016*). It also suggests that the natural expansion in adult zebrafish is regulated in part by physiological demands of the cardiac tissue to respond to the increasing body size of the fish. In much the same way, in juvenile hearts we see a requirement for a second source of blood vasculature as the myocardium expands (*Harrison et al., 2015*). With increased vasculature and heart mass comes increased extravascular fluid and cellular waste products such that the increased interstitial fluid may need an auxiliary conduit from the heart back to the circulation. The cardiac lymphatic system in zebrafish appears to have the capacity to clear fluid and debris from the myocardium (*Figure 5*).

Only moderate lymphangiogenesis is observed after amputation and the majority of the vasculature in the wound site is blood vasculature (*Figure 4*). Our analyses have demonstrated that there is a marked effect on the vascular response injury after extensive tissue damage (cyroinjury). Upon cryoinjury, there is a significant amount of lymphangiogenesis in and around the regeneration site that is not observed after amputation (*Figure 4*). Both injury models lead to increased *vegfc* levels, although this expression is more prolonged after cyroinjury (*Figure 4*). Coupled with a strong immune response to cryoinjury that could utilize the cardiac lymphatic system for the removal of cell debris and immune clearance (*Figure 5*) we postulate that this response can avoid a prolonged inflamed wound site and aid regeneration (*Lai et al., 2017*; *Vieira et al., 2018*). Consistent with this we see large scar tissue volume following severe cryoinjury of zebrafish ventricles that lack lymphatic vessels (*Figure 6*). Scar tissue is detectable in control ventricles at 60dpc, but the size and severity of this scar tissue is less than that observed in *hsp70l:flt4* zebrafish without cardiac lymphatic

vasculature (*Figure 6*). We also observe significantly more neutrophils at the 14dpc wound site with post-injury induction sFlt4 (*Figure 6*). This is early in the response of the lymphatic vessels to injury (*Figure 4*), but suggests compromised ability of the ventricular cardiac lymphatic vessels to remove the influx of neutrophils to the woundsite (*Lai et al., 2017*) with blocking Vegfc-signaling. We cannot rule out a direct effect on elevating the immune response with blockade of Vegfc-signal, but the resulting shift has a potent negative effect on the continuing regenerative environment of the tissue, while not directly effecting cardiomyocyte proliferation, epicardial activation, coronary vessel revascularization or heart structure and morphology (*Figure 6*). Consistent with our studies using sFlt4, recently it has been reported that a *vegfd*-mutation in a *vegfc*-hypermorphic background can also limit the extension of the cardiac lymphatic vessels down the ventricle while not affecting coronary vessels (*Vivien et al., 2019*). Further, almost half of the *vegfc/d* mutant zebrafish surviving to adulthood fail to complete regeneration by 180dpc (*Vivien et al., 2019*). Interestingly, these zebrafish also show cardiac hypertrophy, which we did not observe after induction of *sflt4* at juvenile stages (*Figure 6—figure supplement 1*). It remains to be determined if *sflt4* induction during embryonic stages results in cardiac hypertrophy. Nonetheless our results suggest that disruption of the cardiac lymphatic system impacts the hearts response to injury, rather than the enlargement of the myocardium.

Shifting the regenerative balance in human hearts for patients that have suffered acute myocardial infarct or insult and related sequelae will likely require the priming of a number of component systems in the heart including that which supplies blood to the injured tissue and removes fluid and debris. Key for this is a detailed understanding of the development of these systems and their interaction in a regenerative environment, which will ultimately help us understand what changes to modulate in patients and how this might be done. Zebrafish has provided an excellent system to understand the development of these systems from a deeper evolutionary understanding through to the ability of these systems to regenerate and play important roles in promoting the regenerative response of adult cardiac tissue to injury.

## Materials and methods

### Zebrafish lines

The following zebrafish lines were raised and maintained at Children's Hospital Los Angeles (CHLA) under standard conditions of care (*Aleström et al., 2019*) and with CHLA IACUC oversight. IACUC vetted and prior approved all experimental procedures used in this study.

Tg(fli1a:EGFP)$^{y1}$(*Lawson and Weinstein, 2002*), Tg(−5.1myl7:DsRed2-NLS)f2 (*Mably et al., 2003*), Tg(−6.5kdrl:mCherry)$^{ci5}$(*Proulx et al., 2010*), Tg(−0.8flt1:RFP)$^{hu5333}$ (referred to as *flt1$^{enh}$: tdtomato*) (*Bussmann et al., 2010*), TgBAC(flt4:Citrine)$^{hu7135}$ (referred to as *flt4:YFP*) (*Gordon et al., 2013*), TgBAC(prox1aBAC:KalTA4-4xUAS-ADV.E1b:TagRFP)$^{nim5}$ (referred to as *prox1a:Gal4-UAS: RFP*) (*van Impel et al., 2014*), Tg(ubb:LOX2272-LOXP-RFP-LOX2272-CFP-LOXP-YFP)$^{a132}$ (referred to as *ubb:zebrabow*) (*Pan et al., 2013*), Tg(kdrl:Cre-ERT2)$^{fb13}$(*Zhao et al., 2014*), Tg(flk1:EGFP)$^{s843}$ (referred to as *kdrl:GFP*) (*Jin et al., 2005*), Tg(gata1a:DsRed)$^{sd2}$ (*Traver et al., 2003*), Tg(hsp70l:flt4, cryaa:Cerulean) (*Matsuoka et al., 2016*), Tg(−5.2lyve1b:DsRed)$^{nz101}$, Tg(−5.2lyve1b: EGFP)$^{nz151}$(*Okuda et al., 2012*), TgBAC(cxcr4a:Citrine)$^{mu104}$ (*Harrison et al., 2015*), Tg(stab1: YFP)$^{hu4453}$ (*Hogan et al., 2009a*), Tg(mrc1a:EGFP)$^{y251}$ (*Jung et al., 2017*), Tg(ubb:LOXP-AmCyan-LOXP-ZsYellow)$^{fb5}$ (referred to as *ubb:CSY*) (*Zhou et al., 2011*), Tg(dll4:EGFP)$^{lcr1}$ (*Sacilotto et al., 2013*) transgenic lines have been described previously, as has the *cxcr4a$^{um20}$* mutation (*Bussmann et al., 2011*).

Tg(hsp70l:sflt4, cryaa:Cerulean) juvenile zebrafish were heat-shocked at 39°C for 1 hr 30 min, three times a week from 35d/1mpf to >6 mpf, or four times a week from 71dpf and 92dpf to >6 mpf (or up to the day prior to surgery, with then no heat-shock 21, 60dpc or 30dpa). Sibling zebrafish (both transgenic and non transgenic) were raised at the same density and heat-shocked at the same time. Non-heat-shocked controls born on the same day were maintained at the same density, but not subjected to heat-shock. Post-injury induction was carried out in a similar fashion. For studies performed at 14 and 56dpc fish were first heat-shocked initially for 2 days prior to injury then four times per week after injury. For studies performed at 3dpc, fish were heat-shocked daily for 10 days (seven prior to injury).

Tg(kdrl:mTurquoise) was generated as outlined (*Bussmann and Schulte-Merker, 2011* and unpublished).

## Immunohistochemistry and in situ hybridization

Standard confocal imaging was carried out as described previously (*Harrison et al., 2015*). Whole mount antibody and cleared native fluorescent zebrafish heart tissue was isolated in the same manner but fixed for 2 hr at room temperature in 4% PFA/PBS before being immunolabeled or mounted in 1% agarose for clearing; further details of which included in a forthcoming publication. Whole-mount immunolabeling was carried out on zebrafish hearts following fixation as follows. Hearts were bleached in Dent's bleach (DMSO:$H_2O_2$:methanol, 1:2.5:40) overnight, rinsed in methanol and then fixed overnight in Dent's fixative (DMSO:Methanol 1:4) at 4˚C and washed in PBS + 0.1%Tween (PBST) for 3 to 8 hr the next day. Hearts were then incubated in primary antibody (see *Table 1*) in blocking solution (HIHS:DMSO:PBS, 1:4:15) for 3–5 days. After which they were washed with PBST for 6 hr. Hearts were then incubated secondary antibody (see *Table 1*) in blocking solution overnight, then washed in PBST for 5 hr. Hearts were then mounted in low melting point agarose and imaging was carried out as before (*Harrison et al., 2015*).

Section immunohistochemistry was carried out on de-waxed paraffin sections (Toluene 5 min x2, 100% ethanol 5 min x2, 80% ethanol 5 min, PBS 5 min x2), after antigen retrieval in Unmasking Solution (Vector Laboratories, H-3300). Primary antibody (see *Table 1*) was applied overnight at 4˚C in blocking solution (5% goat serum, 2.5% BSA, 0.3% Triton, 1% DMSO, 0.1% Tween20 in TBS), after incubation for 1 hr at room temperature in this blocking solution. Slides were rinsed in PBST (5 min x3) before and after application of secondary antibody (1 hr room temperature) then imaged using lsm710 and lambda/spectral imaging.

Section in situ hybridization was carried out on de-waxed paraffin sections using RNAscope Multiplex Fluorescent Assay v2 as per manufacturer's instruction (Cat No. 323100, Advanced Cell Diagnostics), *vegfc* probe (Cat No. 528701, Advanced Cell Diagnostics) was visualized with Opal 690 Reagent Pack (PN: FP1497001KT, Akoya Biosciences).

Quantification of Mpx or IB4-positive cell numbers was carried out in ImageJ though unprojected z-planes, only cells within the lymphatic vessel endothelium were counted. ImageJ was used for quantification of Mpx and IB4-cells in and within 100 μm of the woundsite and averaged across at least three images through woundsite. Similarly quantification of percentage PCNA/Mef2c-positive cells was carried by counting double positive and Mef2c-only cells within a 500 μm$^2$ window proximal to the woundsite, and then averaged across at least three images. Quantification of all double positive and Raldh2-only cells within an entire 708 μm$^2$ imaging window proximal to the wound site was carried out to calculate percentage of Raldh2 cells (epicardium) that are proliferative. Quantification of post-injury angiogenic sprouts was carried out by counting *kdrl:GFP*-positive sprouts within the wound site and then dividing this by woundsite area. For *lyve1b:DsRed*-positive, *flt4:mCitrine*-positive lymphatic vessel coverage and *kdrl:GFP*-positive coronary vessel coverage intensity thresholds of a projected z-stack were used to determine the area of the *lyve1*-positive, *flt4:mCitrine*-

**Table 1.** Table of antibodies used.

| Antigen | Species | Concentration | Supplier |
|---|---|---|---|
| Mpx | Rabbit | 1:100 | GeneTex, GTX128379 |
| IB4 (conjugated protein) | n/a (Griffonia Simplicifolia) | 1:100 | Vector Laboratories, DL-1207 |
| GFP | Chicken | 1:500 | Aves Labs, GFP-1010 |
| Rabbit IgG-647 | Goat | 1:500 | ThermoFisher Scientific, A21245 |
| Mouse IgG-488 | Goat | 1:500 | ThermoFisher Scientific, A11001 |
| Chicken IgG-FITC | Goat | 1:500 | Aves Labs, F1005 |
| Raldh2 | Rabbit | 1:200 | GeneTex, GTX124302 |
| Mef2c | Rabbit | 1:200 | Santa Cruz, sc-313 |
| PCNA | Mouse | 1:200 | Vector Labs, VP-P980 |

positive or *kdrl:GFP*-positive vasculature on the ventricle in comparison to the total/partial area of the ventricle or woundsite as defined in text. For ventricle size quantification, the largest AFOG/immune-labeled heart section of each zebrafish was imaged and area measured using threshold function in ImageJ. This maximum area was then plotted against, or normalized to, standard body length of the zebrafish. Raw Mef2c/DAPI counts we made using ImageJ cell counter. Chi-squared statistical analysis of counts was completed in Prism 6 (GraphPad). Visualization and analysis of confocal data were carried out in Zen (Zeiss) and ImageJ, with the exception of lymphatic-neutrophil/macrophage analysis in Vision4D (Arivis).

## Intramyocardial injection

A 10 µl volume of filtered microspheres (0.2 µm Fluoro-Max, B200, Thermo Fisher Scientific) and Qdots (Qtracker705 vascular label, Q21061MP, Thermo Fisher Scientific) was prepared in 50:50 ratio and loaded into a micro-injector pulled-capillary syringe. Zebrafish were anesthetized in Tricaine and the chest opened and held opened with forceps. The injection was made using a FemtoJet microinjector (Eppendorf), 1 µl was injected into the ventral myocardium tissue at an acute angle to the plane of the heart.

## Resection surgery and cryoinjury

Resection surgery and AFOG histology carried out as described previously (*Poss et al., 2002*). Cryoinjury was carried out as described (*González-Rosa and Mercader, 2012*), but with the modification for two injuries: a severe cryoinjury using a 0.8 mm diameter spherical probe and mild cryoinjury using a 0.6 mm dia spherical probe (10160–13, Fine Scientific Tools). Both are cooled in $LN_2$ prior to application to the ventricle for 7–10 s after which the probe was warmed with water and removed, the post-surgery fish were returned to system water to recover. At least five zebrafish were injured per observation/group to allow for potential variability of the wound response within a single experiment. For comparative studies sibling zebrafish were used when possible and randomly assigned into different injury groups.

Estimation of lymphatic vessel coverage after the injury was calculated in a fixed 600 $\mu m^2$ square section centered on the wound site or uninjured apical region. Using thresholding of the 514 nm channel in ImageJ area of the lymphatic vessel was calculate and results presented as percentage coverage of the fixed area of measurement. To quantify the relative level of branching after injury, vessels were traced and total length calculated in ImageJ, then total number of bifurcations over the ventricle counted to give normalized bifurcations per unit length of vessel. For cyroinjury scar volume calculations, whole ventricle and scar (collagen and fibrin) regions were traced as regions of interest in imageJ. Intensity thresholding was used to select tissue to calculate the area without the inclusion of intertrabecular space. This area measurement was repeated on every $9^{th}$ 7 µm consecutive section through the entire heart. The individual area measurements were then multiplied by 63 µm and subsequently added together to give a total estimated volume. Hearts were grouped based on the volume of this scar tissue with a cut off set at 0.05 $\mu m^3$ above which was defined as large scar. For amputation, hearts were grouped on the basis of collagen and fibrin levels. Hearts with or less than a thin layer of collagen (blue) on the inside of the thickened regenerated tissue were classed as 'minimally scared'. Fibrin deposition at the wound site (red) was considered indicative of scar tissue presence. Level of regeneration of the cardiac tissue or level of myocardial wall thickening at the wound site was not considered in scar severity assessments. For quantification of regenerated tissue, thickened myocardial wall proximal to the scar was traced and area calculated for the section though the middle of the scar (largest scar area). For estimation of scar internalization the minimal distance of scar tissue from the edge of the tissue (epicaridum) was measured in ImageJ. Scar area was calculated as volume, but in a single section with the largest scar area.

## Acknowledgements

We thank Drs. Young Hong, Henry Sucov for helpful discussions and suggestions. Drs. Mathias Francois, James Hudson, Ruben Marin-Juez, Enzo Porello and Karina Yaniv for communications prior to the submission. Dr. Elke Ober for sharing the *prox1:RFP* line, Dr. Didier Stainier for the *hsp70l:flt4* line, Dr. Phil Crosier and Dr. Kazuhida Okuda for the *lyve1b* transgenic lines and Dr. Brant Weinstein for the *mrc1a:EGFP* line.

## Additional information

### Funding

| Funder | Grant reference number | Author |
|---|---|---|
| National Heart, Lung, and Blood Institute | 1R01HL130172 | Ching-Ling Lien |
| Children's Hospital Los Angeles | 2nd R01 and Team Awards | Ching-Ling Lien |
| American Heart Association | I81PA34180044 | Ching-Ling Lien |
| California Institute for Regenerative Medicine | EDUC2-08418 | Jessi Villafuerte |
| Deutsche Forschungsgemeinschaft | CRC 1348 | Stefan Schulte-Merker |

The funders had no role in study design, data collection and interpretation, or the decision to submit the work for publication.

### Author contributions

Michael RM Harrison, Conceptualization, Formal analysis, Supervision, Investigation, Visualization, Methodology, Writing—original draft; Xidi Feng, Guqin Mo, Antonio Aguayo, Jessi Villafuerte, Tyler Yoshida, Investigation; Caroline A Pearson, Visualization, Methodology, Writing—review and editing; Stefan Schulte-Merker, Resources, Writing—review and editing; Ching-Ling Lien, Conceptualization, Supervision, Funding acquisition, Methodology, Writing—original draft, Project administration

### Author ORCIDs

Michael RM Harrison https://orcid.org/0000-0003-1703-9879
Stefan Schulte-Merker https://orcid.org/0000-0003-3617-8807
Ching-Ling Lien https://orcid.org/0000-0002-5100-9780

### Ethics

Animal experimentation: All zebrafish husbandry was performed under standard conditions, and all animal experiments were performed in accordance with the recommendations in the Guide for the Care and Use of Laboratory Animals of the National Institutes of Health and following the ARRIVE guidelines provided by the National Centre for the Replacement, Refinement and Reduction of Animals in Research, and the FELASA recommendations on zebrafish husbandry (Alestrom et al., 2019). All procedures were carried out as approved by the Children's Hospital Los Angeles (CHLA)institutional animal care and use committee (IACUC) protocols (#201-18 and 212-16).

### Decision letter and Author response

Decision letter https://doi.org/10.7554/eLife.42762.sa1
Author response https://doi.org/10.7554/eLife.42762.sa2

## Additional files

### Supplementary files

• Transparent reporting form

### Data availability

All data generated or analysed during this study are included in the manuscript and supporting files. Source data files have been provided for Figures 3, 4, 5 and 6.

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
