## [Decision Letter]

Thank you for submitting your article "Late developing cardiac lymphatic vasculature supports adult zebrafish heart function and regeneration" for consideration by *eLife*. Your article has been reviewed by three peer reviewers, one of whom is a member of our Board of Reviewing Editors, and the evaluation has been overseen by a Reviewing Editor and Didier Stainier as the Senior Editor. The following individuals involved in review of your submission have agreed to reveal their identity: Anna Jazwinska (Reviewer #3).

The reviewers have discussed the reviews with one another and the Reviewing Editor has drafted this decision to help you prepare a revised submission.

Summary:

This study by Harrison and colleagues describes the development and response to injury of the cardiac lymphatic vasculature in zebrafish. The authors document the initial formation of lymphatics in the heart at surprisingly late stages, ie in young adults (90dpf), utilising multiple transgenic lines expressing flt4- (vegfr3) and prox1- driven citrine and RFP reporters, and confirmed lymphatic endothelial cell identity with additional lyve-1 and mrc1a reporters plus exclusion of the blood endothelial cell marker kdl. They revealed a close association of the forming lymphatics with coronary blood vessels (arteries) and confirmed similar findings in a comparative study of the lymphatics in human embryonic hearts. They demonstrated that normal coronary vessel formation is a pre-requisite for the extension of the lymphatics from the bulbus cordis down into the ventricle by studying cxcr4 mutant fish with impaired coronaries. The authors then went on to map the response of the lymphatics to heart injury and during regeneration; they initially confirmed the angiogenic response of the coronary blood vasculature, as previously reported, and determined that lymphatics did not sprout into the wound area following ventricular amputation. In contrast, they observed a marked increase in lymphatics around the wound site following cryo-injury suggesting a requirement for necrosis and scarring to induce lymphangiogenesis. Functionally the newly formed lymphatics appeared to take up quantum dots, suggesting fluid clearance arising from the injury may be one of the lymphatic functions and there was a close association with neutrophils and macrophages, suggesting the lymphatics may also traffic immune cells as has recently been shown in mouse. Finally, the authors tested a requirement for vegfc in transgenic fish, with heat-shock induction of soluble flt4/vegfr3 receptor, which led to significantly fewer LECs, no lymphatics within the ventricle and correlated with increased severity of scarring in cryo-injured fish.

This is an interesting and novel study, as the first to describe the development and response to injury of the cardiac lymphatics in zebrafish. The data, and in particularly the cleared tissue/whole mount imaging, is of a high standard. The findings are timely, given the recent significant interest in the development and injury-response of organ-based lymphatics across model organisms, as illustrated by several high impact studies. In particular, understanding the response of the cardiac lymphatics to injury is clinically important and the benefit of utilising the zebrafish here is in determining a requirement for lymphatics to facilitate scar resolution and heart regeneration.

That said, there are aspects of the study which have not been developed in sufficient depth to fully support the conclusions and several issues which should be addressed:

Essential revisions:

1) Regarding the relationship of the emerging lymphatics with the coronary vessels, it was not clear whether these were coronary arteries or veins from the images/marker sets utilized. This needs clarifying.

2) The study of human embryos appears somewhat preliminary and superficial relative to the rest of the detailed focus on zebrafish. A suggestion would be to omit the human data here and develop this as a separate study.

3) In the cxcr4 mutant studies, coronary vessels are mis-patterned but clearly still form, as demonstrated by Fli1a staining in Figure 3B. The authors should quantify the density of coronary vessels in these cxcr4a mutants and determine their identity (arterial or venous?). The complete absence of any lymphatics emerging into the ventricle is surprising given the apparent incidence of coronaries. This may reflect the fact that cxcr4 signalling is directly required for LEC guidance as opposed to acting through the coronary vessels. At present the suggestion that the coronaries act as a guide/scaffold for the lymphatics is only correlative; statements such as "lymphatic development requires a coronary artery scaffold" and in the abstract "Zebrafish cardiac lymphatic vessels require coronary arteries to guide their expansion" are not supported by the data. These need to be toned down or further supported by functional data showing either that coronaries are sufficient to provide guidance or that cxcr4 is in fact required more directly to guide the forming lymphatics.

4) Figure 4, the analysis of the angiogenic response to injury is essentially confirming prior work (Marin-Juez et al., 2016) and is largely redundant here. It would be better to focus on the lymphatics, albeit alongside blood vessels, in response to injury. A suggestion would be to omit Figure 4, and the corresponding supplementary figure, and to focus on the lymphatics with further images of sprouting lymphatics into the wound area (alongside the blood vasculature).

5) Some additional analyses of the responding lymphatics would be highly informative, to determine whether the newly forming vessels are different (marker profile, anatomy, organisation) from the pre-existing lymphatics.

6) A more detailed time course of the lymphangiogenic response to injury is required. Analysis at a single time point, of 54 dpa or 60 dpc, as presented fails to address important questions such as, when do the lymphatics first respond and/or is there a transient or delayed response following resection (amputation)?

7) The functional studies in Figure 6 are somewhat premature and insufficient to support the conclusions. The presence of Qdots within lymphatics needs demonstrating unequivocally by improved orthogonal imaging to assess whether the dots are actually within the vessel lumen. In addition, a demonstration of clearance of Qdots to sites outside of the heart (analogous to draining lymph nodes in mammals) would be informative regarding function. If these additional analyses are deemed challenging. then it would be preferable not to include the Qdot data and to omit conclusions on attributing a clearance function to the cardiac lymphatics.

8) For the immune cell studies, it is not clear how the authors can claim that neutrophils are in lymphatic vessels given the reliance on Fli1a:GFP as a marker (Figure 6G); this labels all endothelial cells and ought to be replaced with a lymphatic-specific reporter. The evidence that Mpx+ neutrophils are actually within lymphatic vessels requires more detailed orthogonal imaging and other important immune cell types should be assessed, most notably macrophages which dominate the wound healing response to injury can be marked by Mpeg. Improved imaging of both Mpx+ and Mpeg+ cells should be carried out, with quantification of cell numbers that are correlated with the extent of lymphangiogenesis. A suggestion for optimal quantification is to apply FACS, making use of available Mpx and Mpeg reporter fish (Renshaw et al., 2006; Ellet et al., 2011).

9) In the inducible soluble flt4 (sFlt4)/vegfr3 transgenic fish the authors should analyse the mutant heart phenotype(s). Do these hearts have any local oedema? Are they malformed? Are the blood vessels of the coronary vasculature quantitatively normal (they look abnormal or different from controls in the images provided)? A careful and thorough description of these hearts is needed because the reduced ability to recover from injury may be secondary to phenotypes caused by the earlier absence of lymphatics, or due to other roles for Flt4 on blood vessels, rather than an acute requirement for lymphatic function during repair. In addition, the lack of lymphangiogenesis following cryoinjury in the sFlt4 mutants suggests that vegfc is probably driving the response; however, there is no expression data shown for vegfc or other lymphangiogenic factors following injury. The authors should analyse vegfc mRNA levels (by combined qPCR and section in situ hybridisation) following cryo-injury versus resection. If vegfc expression is increased following cry-injury, but not following resection, this may explain the difference in lymphatic response between the two injury models.

Finally, heat-shock, regardless of carrying the sFlt4 transgene, appears to have reduced coronary blood vessels (Figure 7A-C). Quantification of the blood vessels should be provided across experimental groups, as the reduced regeneration could be due to a blood vascular phenotype and may be related to heat shock rather than the flt4/vegfc signalling. Why do the authors think the heat-shocked controls mounted a lymphatic response? This should be commented on further in the text.

10) The assessment of the effect of sFlt4 (impaired lymphangiogenesis) on scarring, as shown in Figure 7, requires a time course to determine whether the effect is with respect to initial scar deposition versus subsequent scar resolution and how this relates to regeneration per se. What are the effects of sFlt4 on other cell types and regenerative processes, such as activation of epicardium/endocardium and cardiomyocyte proliferation? Moreover, to link to the earlier functional studies the authors should investigate an effect on immune cell numbers (Mpx+ and Mpeg+ cells) in the absence of lymphatics.

[Editors' note: further revisions were requested prior to acceptance, as described below.]

Thank you for resubmitting your work entitled "Late developing cardiac lymphatic vasculature supports adult zebrafish heart function and regeneration" for further consideration at *eLife*. Your revised article has been favorably evaluated by Didier Stainier (Senior Editor), a Reviewing Editor, and two reviewers.

The manuscript has been significantly improved, but there are a couple of remaining issues that need to be addressed before acceptance, as outlined below:

1) Some further data is required to demonstrate the specificity of the cardiac phenotype following the transgenic loss of lymphatics after injury and to establish rigorously that the cardiac lymphatics are required for heart regeneration. In the sFlt4 model the coronary blood vasculature and general heart morphology should be investigated post-injury, to determine whether they are relatively normal, and thus place the emphasis more specifically on the loss-of-lymphatics.

2) The role of the lymphatics in immune cell trafficking and specifically the data on neutrophils is correlative at this stage. The images in Figure 5 and Video 2 are not entirely convincing, as to whether the documented Mpx+ cells actually reside within the vessel lumen, and there is no functional validation such as via Mtz-mediated neutrophil depletion. Whilst the latter is recognised to be beyond the current scope, and would require several months of new experimentation, there is a need to tone down statements throughout on immune cell/neutrophil clearance and especially with regards being causative in regeneration versus scarring.

---

## [Author Response]

Essential revisions:1) Regarding the relationship of the emerging lymphatics with the coronary vessels, it was not clear whether these were coronary arteries or veins from the images/marker sets utilized. This needs clarifying.

Thank you for pointing out that this is not to sufficiently clear. We have added additional analysis of markers to help address this. We have shown lymphatic formation along vessels expressing dll4^high^ (Figure 2) cxcr4a (Figure 3), flt1^enh^ (Figure 2) and Kdrl^high^ (Figure 2). We also show that cxcr4a overlaps with flt1^enh^ expression (Figure 2) and have previously shown this expression overlaps with dll4^high^ and is consistent with the expression of these markers in thearteries of the fins (Harrison et al., 2015).

2) The study of human embryos appears somewhat preliminary and superficial relative to the rest of the detailed focus on zebrafish. A suggestion would be to omit the human data here and develop this as a separate study.

We agree with the suggestion and have removed the human embryo data at this stage.

3) In the cxcr4 mutant studies, coronary vessels are mis-patterned but clearly still form, as demonstrated by Fli1a staining in Figure 3B. The authors should quantify the density of coronary vessels in these cxcr4a mutants and determine their identity (arterial or venous?). The complete absence of any lymphatics emerging into the ventricle is surprising given the apparent incidence of coronaries. This may reflect the fact that cxcr4 signalling is directly required for LEC guidance as opposed to acting through the coronary vessels. At present the suggestion that the coronaries act as a guide/scaffold for the lymphatics is only correlative; statements such as "lymphatic development requires a coronary artery scaffold" and in the abstract "Zebrafish cardiac lymphatic vessels require coronary arteries to guide their expansion" are not supported by the data. These need to be toned down or further supported by functional data showing either that coronaries are sufficient to provide guidance or that cxcr4 is in fact required more directly to guide the forming lymphatics.

We agree with this comment, and have accordingly toned down the use of “require” (subsection “Development of a cardiac lymphatic system is incomplete in the absence of a coronary artery scaffold”). We are also interested in the potential role of cxc-signaling in the extension of the lymphatics and their association with arteries and agree that this is still a possibility. The fact that this signaling could be more broadly affected in the mutant has been noted in the text regardless of any effect on the lymphatic endothelial cells directly (Discussion section).

We have looked in more detail at the correlation of between kdrl^high^ and lymphatic marker expression/coverage, both of which vary within the WT population and between heart sides within individual zebrafish (Figure 3—figure supplement 2). There is a highly significant correlation suggesting the former is required or promotes the extension of lymphatics, but we agree this is a suggestion that needs more independent verification and have reflected this in the text (subsection “Development of a cardiac lymphatic system is incomplete in the absence of a coronary artery scaffold”).

We are also currently looking at other markers of coronary vessel identity in the cxcr4a mutant background, but this work could not be completed within the revision timeframe. However, we have studied higher numbers of mutant and control fish and now provide more quantification of these vessels and the variation observed in older cxcr4a mutant fish (Figure 3—figure supplement 2). We believe this clarifies the point we are making with respect to abnormal (arterial) coronary development, while not precluding a more direct role for cxc-signaling in lymphatic development.

4) Figure 4, the analysis of the angiogenic response to injury is essentially confirming prior work (Marin-Juez et al., 2016) and is largely redundant here. It would be better to focus on the lymphatics, albeit alongside blood vessels, in response to injury. A suggestion would be to omit Figure 4, and the corresponding supplementary figure, and to focus on the lymphatics with further images of sprouting lymphatics into the wound area (alongside the blood vasculature).

We have removed the coronary data and instead focused on the lymphatogenesis response in particular the marker profile (5) and time course (6).

5) Some additional analyses of the responding lymphatics would be highly informative, to determine whether the newly forming vessels are different (marker profile, anatomy, organisation) from the pre-existing lymphatics.

We have characterized the expression of Lyve1:DsRed, Flt4:mCitrene, Mrc1a:GFP, kdrl:mTurquiose and Prox1:RFP in regenerated lymphatic vessels and quantified the branching organization observed in Flt4:mCitrene fish (Figure 4). We now also provide a better description of the responding lymphatics.

6) A more detailed time course of the lymphangiogenic response to injury is required. Analysis at a single time point, of 54 dpa or 60 dpc, as presented fails to address important questions such as, when do the lymphatics first respond and/or is there a transient or delayed response following resection (amputation)?

We have analyzed the lymphangiogenesis response earlier with both injury models finding no such transient response to amputation and overt lymphangiogenesis observed by 28dpc, with more subtle changes observed earlier (Figure 4).

7) The functional studies in Figure 6 are somewhat premature and insufficient to support the conclusions. The presence of Qdots within lymphatics needs demonstrating unequivocally by improved orthogonal imaging to assess whether the dots are actually within the vessel lumen. In addition, a demonstration of clearance of Qdots to sites outside of the heart (analogous to draining lymph nodes in mammals) would be informative regarding function. If these additional analyses are deemed challenging. then it would be preferable not to include the Qdot data and to omit conclusions on attributing a clearance function to the cardiac lymphatics.

We agree that showing clearance to remote sites outside the heart would be informative and we thank the reviewers for the suggestion, however given the uncertain anatomy this has proved difficult for this current study.

It appears that flow in the cardiac lymphatic vessel is up the aorta and joins collection from the gills (Figure 2—figure supplement 1) to extend toward the facial lymphatics. However, imaging this extracardiac region is difficult and so we instead decided to remove this analysis, but include clearer imaging of earlier uptake of the Qdots on the ventricle. In doing so we make a simpler point that the ventricular lymphatics form a functional vessel capable of Qdot uptake (Figure 5). We have made the necessary changes to the text to reflect this conclusion subsection “Cardiac lymphatic vasculature functionally supports the heart during regeneration and homeostasis.” and Discussion section).

8) For the immune cell studies, it is not clear how the authors can claim that neutrophils are in lymphatic vessels given the reliance on Fli1a:GFP as a marker (Figure 6G); this labels all endothelial cells and ought to be replaced with a lymphatic-specific reporter. The evidence that Mpx+ neutrophils are actually within lymphatic vessels requires more detailed orthogonal imaging and other important immune cell types should be assessed, most notably macrophages which dominate the wound healing response to injury can be marked by Mpeg. Improved imaging of both Mpx+ and Mpeg+ cells should be carried out, with quantification of cell numbers that are correlated with the extent of lymphangiogenesis. A suggestion for optimal quantification is to apply FACS, making use of available Mpx and Mpeg reporter fish (Renshaw et al., 2006; Ellet et al., 2011).

Lymphatic vessels we identified based on morphology and position, but we agree this is not ideal and so have repeated the experiment with Lyve1GFP zebrafish and in addition carried out higher resolution orthogonal imaging (Figure 5).

Unfortunately we do not have the suggested Mpx or Mpeg zebrafish in our stocks to carry out the experiment as suggested and crossing these fish to lymphatic reporter lines will need more time than the revision allows, but we did successfully used Isolectin B4 conjugate to identify macrophage population in wholemount hearts and included this analysis. We used both markers to determine if there was a prolonged or heightened immune response in the absence of cardiac lymphatic extension/post-injury lymphatogenesis, finding a significant increase of Mpx-cells at the 14dpc wound site (Figure 6), an elevated but not significant increases later at 56dpc (post-injury, Figure 6—figure supplement 1) and 60dpc (developmental extension, Figure 6—figure supplement 1). We have discussed the implications of this (Discussion section) in the context of lymphatic function.

9) In the inducible soluble flt4 (sFlt4)/vegfr3 transgenic fish the authors should analyse the mutant heart phenotype(s). Do these hearts have any local oedema? Are they malformed? Are the blood vessels of the coronary vasculature quantitatively normal (they look abnormal or different from controls in the images provided)? A careful and thorough description of these hearts is needed because the reduced ability to recover from injury may be secondary to phenotypes caused by the earlier absence of lymphatics, or due to other roles for Flt4 on blood vessels, rather than an acute requirement for lymphatic function during repair. In addition, the lack of lymphangiogenesis following cryoinjury in the sFlt4 mutants suggests that vegfc is probably driving the response; however, there is no expression data shown for vegfc or other lymphangiogenic factors following injury. The authors should analyse vegfc mRNA levels (by combined qPCR and section in situ hybridisation) following cryo-injury versus resection. If vegfc expression is increased following cry-injury, but not following resection, this may explain the difference in lymphatic response between the two injury models.Finally, heat-shock, regardless of carrying the sFlt4 transgene, appears to have reduced coronary blood vessels (Figure 7A-C). Quantification of the blood vessels should be provided across experimental groups, as the reduced regeneration could be due to a blood vascular phenotype and may be related to heat shock rather than the flt4/vegfc signalling. Why do the authors think the heat-shocked controls mounted a lymphatic response? This should be commented on further in the text.

The reviewers make a number of valid points here. Interestingly we did not find overt edema at 6 months, but it is possible this develops in older adults – we have commented on this and the histology of the sFlt4 hearts (subsection “Vegfc promotes the formation of cardiac lymphatic vasculature after the initiation of coronary vessel development”).

We agree that any effect on coronary vessels should be analyzed more closely. We see only transient expression of Flt4 in the angiogenic sprouts emerging from the AVC, but not after when they extend over the ventricle and when sFlt4 induction was initiated. We have included here that developmental induction from 1mpf (after initial coronary vessel development had begun) appears to not overtly affect the kdrl+ coronary vasculature (Figure 6—figure supplement 1) and certainly not the arterial structure that we believe vital for ventricle lymphatic vessel extension.

We have tested if vegfc is driving the lymphangiogenesis response, both by analyzing expression and inducing sFlt4 specifically after injury (Figure 4, see point 10 below).

We agree that it is interesting that heat shocked zebrafish mount a lymphatic response. It is likely in response to the stress placed on heart during heat shock, including increased heart output and the repercussions of this on myocardial energy/waste production. We have discussed this further in the text and think it is an interesting avenue of investigation (Discussion section).

10) The assessment of the effect of sFlt4 (impaired lymphangiogenesis) on scarring, as shown in Figure 7, requires a time course to determine whether the effect is with respect to initial scar deposition versus subsequent scar resolution and how this relates to regeneration per se. What are the effects of sFlt4 on other cell types and regenerative processes, such as activation of epicardium/endocardium and cardiomyocyte proliferation? Moreover, to link to the earlier functional studies the authors should investigate an effect on immune cell numbers (Mpx+ and Mpeg+ cells) in the absence of lymphatics.

We agree that a better understanding of scar formation and removal and examining other processes of heart regeneration in animals with forced sflt4 expression would be beneficial; we have raised more developmentally induced-sflt4 zebrafish to investigate this further in the future. However, raising these fish took 6 months and the length subsequent post-injury analysis has prevented this work from completion in the revision time period. We instead followed reviewer’s suggestion that post-injury blockade of VEGFC-signal may cause a lack of lymphangiogenesis following cryoinjury in the sFlt4 fish. We have found that post-injury induction sflt4 to block the lymphangiogenic response does inhibit regeneration, but such effects are more subtle (Figure 6—figure supplement 2). In addition, we see no difference in scar size (or tissue regeneration) in sFlt4induced fish at 14dpc, suggesting collagen deposition is not elevated.

We have evaluated the effect of sFlt4 induction on cardiomyocyte proliferation and activation of the epicardium (Figure 6—figure supplement 3). Further we have quantified immune cell number at the wound site in both post-injury and developmentally induced fish to observe is prolonged elevated inflammation in these fish (Figure 6—figure supplement 1 and Figure 6—figure supplement 2). We think this provides an interesting and informative complement to the data previously submitted and have discussed these trends more fully in the text (Discussion section).

[Editors' note: further revisions were requested prior to acceptance, as described below.]

The manuscript has been significantly improved, but there are a couple of remaining issues that need to be addressed before acceptance, as outlined below:1) Some further data is required to demonstrate the specificity of the cardiac phenotype following the transgenic loss of lymphatics after injury and to establish rigorously that the cardiac lymphatics are required for heart regeneration. In the sFlt4 model the coronary blood vasculature and general heart morphology should be investigated post-injury, to determine whether they are relatively normal, and thus place the emphasis more specifically on the loss-of-lymphatics.

We have examined and/or quantified the cardiac morphology and coronary vessels using both developmentally induced and post-injury induced sFlt4. We did not find significant edema or cardiac hypertrophy and have included more data and quantification to bolster this.

Developmental: whole-mount brightfield imaging, ventricle size, cardiomyocytes and noncardiomyocytes analysis (Fig6—figure supplement 1).

Post-injury: histology and post-injury ventricle area to body length comparisons (Fig6—figure supplement 3),

In addition to finding the coronary vessels are largely normal with developmental induction of sFlt4 (Figure 6—figure supplement 1) we find that injury of these vessels, or post-injury induction of sFlt4, does not significantly attenuate the coronary vessel response to injury (at 14 and 21dpc, Fig6—figure supplement 2 and Figure 6—figure supplement 3). Together this suggests that sFlt4 overexpression does not affect angiogenesis of coronary vessels.

Indeed, for sFlt4 induction during development, we have induced the transgene at 71 and 92dpf (Fig6—figure supplement 1) to ensure that sFlt4 is only over-expressed late in or after the completion of coronary vessel development and this is still sufficient to strongly inhibit the formation of the cardiac lymphatic vessel. This further suggests the observed sFlt4 phenotype is indeed a specific effect on the flt4-expressing cardiac lymphatic endothelial cells and not explained by a repercussion of sFlt4 induction affecting the coronary vasculature.

This finding is consistent with a recent publication by Vivien et al., in npj Regenerative Medicine.

2) The role of the lymphatics in immune cell trafficking and specifically the data on neutrophils is correlative at this stage. The images in Figure 5 and Video 2 are not entirely convincing, as to whether the documented Mpx+ cells actually reside within the vessel lumen, and there is no functional validation such as via Mtz-mediated neutrophil depletion. Whilst the latter is recognised to be beyond the current scope, and would require several months of new experimentation, there is a need to tone down statements throughout on immune cell/neutrophil clearance and especially with regards being causative in regeneration versus scarring.

We agree this imaging is less than convincing and have provided new imaging data to demonstrate that mpx+ neutrophils reside within the vessel lumen (Figure 5—figure supplement 1, Figure 5Video 4 and Video 5). However, we agree with the reviewers that we observed more macrophages than neutrophils clearly residing within the lumen of the lymphatic vessels, while some of the neutrophils maybe along or enveloped within the vessel membrane. Therefore, we have revised the results as “mpx+ neutrophils associated with cardiac lymphatic vessels” (Figure 5—figure supplement 1) and toned down the conclusions relating to this throughout the manuscript (Abstract, Introduction, subsection “Cardiac lymphatic vasculature functionally supports the heart during regeneration and homeostasis.”, subsection “Vegfc promotes the formation of cardiac lymphatic vasculature after the initiation of coronary vessel development”, Discussion section and subsection “Zebrafish lines”).